# LVSM: A Large View Synthesis Model with Minimal 3D Inductive Bias

**Haian Jin**[1]   **Hanwen Jiang**[2]   **Hao Tan**[3]   **Kai Zhang**[3]   **Sai Bi**[3]   **Tianyuan Zhang**[4]
**Fujun Luan**[3]   **Noah Snavely**[1]   **Zexiang Xu**[3]
[1]Cornell University  [2]The University of Texas at Austin
[3]Adobe Research  [4]Massachusetts Institute of Technology

## Abstract

We propose the Large View Synthesis Model (LVSM), a novel transformer-based approach for scalable and generalizable novel view synthesis from sparse-view inputs. We introduce two architectures: (1) an encoder-decoder LVSM, which encodes input image tokens into a fixed number of 1D latent tokens, functioning as a fully learned scene representation, and decodes novel-view images from them; and (2) a decoder-only LVSM, which directly maps input images to novel-view outputs, completely eliminating intermediate scene representations. Both models bypass the 3D inductive biases used in previous methods—from 3D representations (e.g., NeRF, 3DGS) to network designs (e.g., epipolar projections, plane sweeps)—addressing novel view synthesis with a fully data-driven approach. While the encoder-decoder model offers faster inference due to its independent latent representation, the decoder-only LVSM achieves superior quality, scalability, and zero-shot generalization, outperforming previous state-of-the-art methods by 1.5 to 3.5 dB PSNR. Comprehensive evaluations across multiple datasets demonstrate that both LVSM variants achieve state-of-the-art novel view synthesis quality. Notably, our models surpass all previous methods even with reduced computational resources (1-2 GPUs). Please see our website for more results: https://haian-jin.github.io/projects/LVSM/.

## 1 Introduction

Novel view synthesis is a long-standing challenge in vision and graphics. For decades, the community has generally relied on various 3D inductive biases, incorporating 3D priors and handcrafted structures to simplify the task and improve synthesis quality. Recently, NeRF, 3D Gaussian Splatting (3DGS), and their variants (Mildenhall et al., 2020; Barron et al., 2021; Müller et al., 2022; Chen et al., 2022; Xu et al., 2022; Kerbl et al., 2023; Yu et al., 2024b) have significantly advanced the field by introducing new inductive biases through carefully designed 3D representations (e.g., continuous volumetric fields and Gaussian primitives) and rendering equations (e.g., ray marching and splatting with alpha blending), reframing view synthesis as the optimization of the representations using rendering losses on a per-scene basis. Other methods have also built generalizable networks to estimate these representations or directly generate novel-view images in a feed-forward manner, often incorporating additional 3D inductive biases, such as projective epipolar lines or plane-sweep volumes, in their architecture designs (Wang et al., 2021a; Yu et al., 2021; Chen et al., 2021; Suhail et al., 2022b; Charatan et al., 2024; Chen et al., 2024).

While effective, these 3D inductive biases inherently limit model flexibility, constraining their adaptability to more diverse and complex scenarios that do not align with predefined priors or handcrafted structures. Recent large reconstruction models (LRMs) (Hong et al., 2024; Li et al., 2023; Wei et al., 2024; Zhang et al., 2024) have made notable progress in removing architecture-level biases by leveraging large transformers without relying on epipolar projections or plane-sweep volumes, achieving state-of-the-art novel view synthesis quality. However, despite these advances, LRMs still rely on representation-level biases—such as NeRFs, meshes, or 3DGS, along with their respective rendering equations—that limit their potential generalization and scalability.

In this work, we aim to *minimize 3D inductive biases* and push the boundaries of novel view synthesis with a fully data-driven approach. While some prior works have attempted to reduce 3D inductive biases (Sitzmann et al., 2021; Sajjadi et al., 2022; Kulhánek et al., 2022), they still face challenges

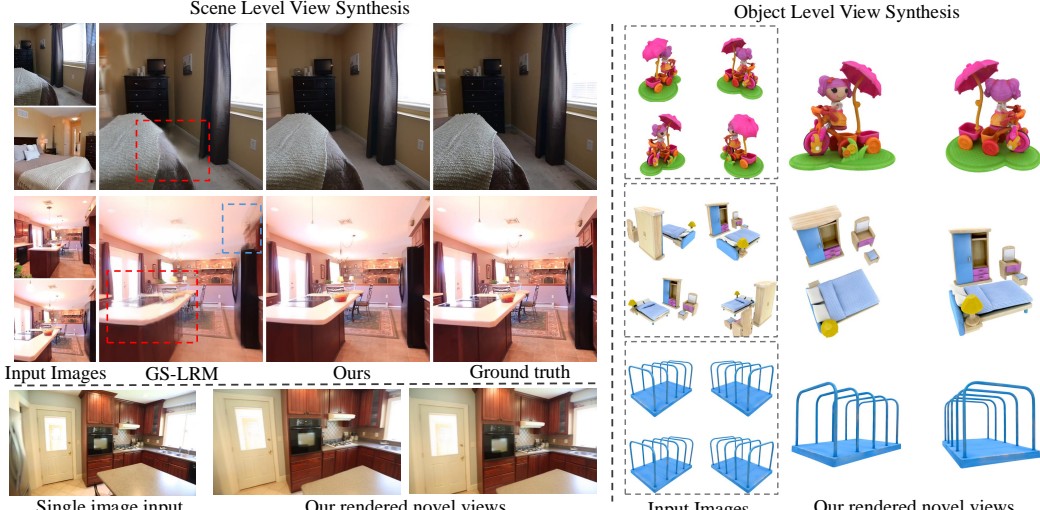

Scene Level View Synthesis

Object Level View Synthesis

Input Images  GS-LRM  Ours  Ground truth

Single image input  Our rendered novel views

Input Images  Our rendered novel views

Figure 1: LVSM supports feed-forward novel view synthesis from sparse posed image inputs (even from a single view) on both objects and scenes. LVSM achieves significant quality improvements compared with the previous SOTA method, i.e., GS-LRM (Zhang et al., 2024). (Please zoom in for more details.)

in scalability and achieving high rendering quality. We propose the **Large View Synthesis Model (LVSM)**, a novel transformer-based framework that synthesizes novel-view images from posed sparse-view inputs *without predefined rendering equations or 3D structures, enabling accurate, training-efficient, and scalable novel view synthesis with photo-realistic quality* (as shown in Fig. 1).

To this end, we first introduce an **encoder-decoder LVSM**, removing handcrafted 3D representations and their rendering equations. We use an encoder transformer to map the input (patchified) multi-view image tokens into a fixed number of 1D latent tokens, independent of the number of input views. These latent tokens are then processed by a decoder transformer, which uses target-view Plücker rays as positional embeddings to generate the target view's image tokens, ultimately regressing the output pixel colors from a final linear layer. The encoder-decoder LVSM jointly learns a reconstructor (encoder), a scene representation (latent tokens), and a renderer (decoder) directly from data. By removing the need for predefined inductive biases in rendering and representation, LVSM offers improved generalization and achieves higher quality compared to NeRF- and GS-based approaches.

Nevertheless, the encoder-decoder LVSM retains a key inductive bias: the use of an intermediate, albeit fully learned, scene representation. To further push the boundary, we propose a **decoder-only LVSM**, which adopts a single-stream transformer to directly convert input view tokens and target pose tokens into target view tokens, bypassing any intermediate representations. The decoder-only LVSM integrates novel view synthesis process into a holistic data-driven framework, achieving simultaneous scene reconstruction and rendering in a fully implicit manner with minimal 3D inductive bias.

We present a comprehensive evaluation of variants of both LVSM architectures. Notably, our models, trained on 2 or 4 input views, demonstrate strong zero-shot generalization to an unseen number of views, ranging from a single input to more than 10. Thanks to minimal inductive biases, our decoder-only model consistently outperforms the encoder-decoder variant in terms of quality, scalability, and zero-shot capability with varying numbers of input views. On the other hand, the encoder-decoder model achieves much faster inference speed due to its use of a fixed-length latent scene representation. Both models, benefiting from reduced 3D inductive biases, outperform previous methods, achieving state-of-the-art novel view synthesis quality across multiple object-level and scene-level benchmark datasets. Specifically, **our decoder-only LVSM surpasses previous state-of-the-art methods, such as GS-LRM, by a substantial margin of 1.5 to 3.5 dB PSNR**. Our final models were trained on 64 A100 GPUs for 3-7 days, depending on the data type and model architecture, but we found that even with just 1–2 A100 GPUs for training, our model (with a decreased model and batch size) still outperforms all previous methods trained with equal or even more compute resources.

## 2 RELATED WORK

**View Synthesis.** Novel view synthesis (NVS) has been studied for decades. Image-based rendering (IBR) methods perform view synthesis by weighted blending of input reference images using proxy

geometry (Debevec et al., 1996; Heigl et al., 1999; Sinha et al., 2009). Light field methods build a slice of the 4D plenoptic function from dense view inputs (Gortler et al., 1996; Levoy & Hanrahan, 1996; Davis et al., 2012). Recent learning-based IBR methods incorporate convolutional networks to predict blending weights (Hedman et al., 2018; Zhou et al., 2016; 2018) or use predicted depth maps (Choi et al., 2019). However, the renderable region is usually constrained to be near the input views. Other work leverages multiview-stereo reconstructions to render larger viewpoint changes (Jancosek & Pajdla, 2011; Chaurasia et al., 2013; Penner & Zhang, 2017). In contrast, we use more scalable network designs to learn generalizable priors from larger, real-world data. Moreover, we perform rendering at the image patch level, achieving better model efficiency and rendering quality.

**Optimizing 3D Representations.** NeRF (Mildenhall et al., 2020) introduced a neural volumetric 3D representation with differentiable volume rendering, enabling neural scene reconstruction by minimizing rendering losses and setting a new standard in NVS. Later work improved NeRF with better rendering quality (Barron et al., 2021; Verbin et al., 2022; Barron et al., 2023), faster optimization or rendering speed (Reiser et al., 2021; Hedman et al., 2021; Reiser et al., 2023), and looser requirements on the input views (Niemeyer et al., 2022; Martin-Brualla et al., 2021; Wang et al., 2021b). Other work has explored hybrid representations that combine implicit NeRF content with explicit 3D information, e.g., in the form of voxels, as in DVGO (Sun et al., 2022). Spatial complexity can be further decreased by using sparse voxels (Liu et al., 2020; Fridovich-Keil et al., 2022), volume decomposition (Chan et al., 2022; Chen et al., 2022; 2023), and hashing techniques (Müller et al., 2022). Another line of work investigates explicit point-based representations (Xu et al., 2022; Zhang et al., 2022; Feng et al., 2022). Gaussian Splatting (Kerbl et al., 2023) extends these 3D points to 3D Gaussians, improving both rendering quality and speed. In contrast, we perform NVS using large transformer models (optionally with a learned latent scene representation) without requiring the inductive bias of using prior 3D representations, nor any per-scene optimization process.

**Generalizable View Synthesis Methods.** Generalizable methods enable fast NVS inference by using neural networks, trained across scenes, to predict novel views or an underlying 3D representation in a feed-forward manner. For example, PixelNeRF (Yu et al., 2021), MVSNeRF (Chen et al., 2021) and IBRNet (Wang et al., 2021a) predict volumetric 3D representations from input views, utilizing 3D-specific priors like epipolar lines or plane sweep cost volumes. Later methods improve performance under (unposed) sparse views (Liu et al., 2022; Johari et al., 2022; Jiang et al., 2024; Szymanowicz et al., 2024b; Jiang et al., 2023), while other work extends to 3DGS representations Charatan et al. (2024); Szymanowicz et al. (2024a); Chen et al. (2024); Tang et al. (2024). On the other hand, approaches that attempt to directly learn a geometry-free rendering function (Suhail et al., 2022a; Sajjadi et al., 2022; Sitzmann et al., 2021; Rombach et al., 2021; Kulhánek et al., 2022) lack model capacity and scalability, preventing them from capturing high-frequency detail. Specifically, Scene Representation Transformers (SRT) (Sajjadi et al., 2022) also aims to avoid explicit, handcrafted 3D representations and instead learns a latent representation via a transformer, similar to our encoder-decoder model. However, some of SRT's model and rendering designs lead to less effective performance, such as the CNN-based token extractor and the use of cross-attention in the decoder. In contrast, our models are fully transformer-based with bidirectional self-attention (see detailed discussion in Sec. 4.4). Additionally, we introduce a more scalable decoder-only architecture that effectively learns the NVS function with minimal 3D inductive bias, without an intermediate latent representation.

Recently, 3D large reconstruction models (LRMs) have emerged (Hong et al., 2024; Li et al., 2023; Wang et al., 2023a; Xu et al., 2023; Wei et al., 2024; Zhang et al., 2024; Xie et al., 2024), utilizing scalable transformer architectures (Vaswani et al., 2017) trained on large datasets to learn generic 3D priors. While these methods avoid using epipolar projection or cost volumes in their architectures, they still rely on existing 3D representations like tri-plane NeRFs, meshes, or 3DGS, along with their corresponding rendering equations, limiting their flexibility. In contrast, our approach eliminates these 3D inductive biases, aiming to learn a rendering function (and optionally a scene representation) directly from data. This leads to more scalable models and significantly improved rendering quality.

In addition to the deterministic methods mentioned above, recent generative NVS models support NVS using image/video diffusion models (Watson et al., 2022; Liu et al., 2023a; Gao* et al., 2024; Zheng & Vedaldi, 2024; Kong et al., 2024; Voleti et al., 2025). Note that our LVSM models are deterministic, and thus are fundamentally different from these generative models. We discuss this in detail in the Appendix. A.6.

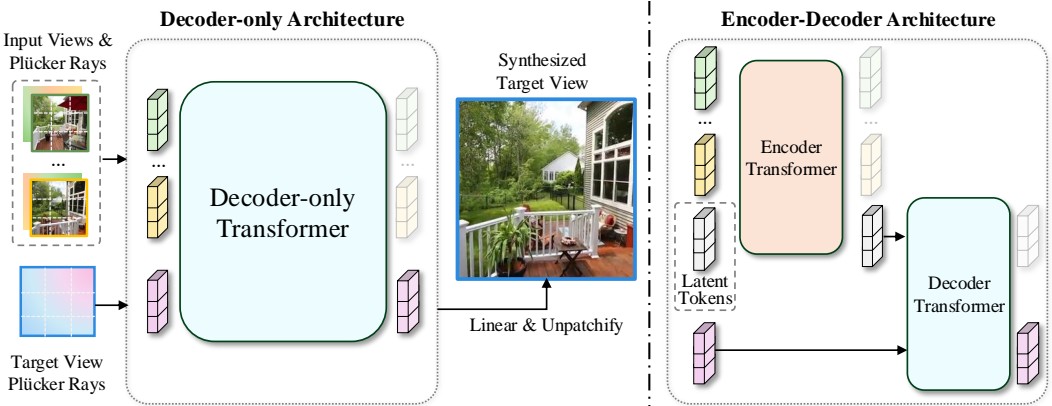

Figure 2: **LVSM model architecture.** LVSM first patchifies the posed input images into tokens. The target view to be synthesized is represented by its Plücker ray embeddings and is also tokenized. The input view and target tokens are sent to a full transformer-based model to predict tokens that are used to regress the target view pixels. We study two LVSM transformer architectures, a *Decoder-only* architecture (left) and a *Encoder-Decoder* architecture (right).

## 3 METHOD

We first provide an overview of our method in Sec. 3.1, then describe two different transformer-based model variants in Sec. 3.2. We also provide detailed architectural diagrams in Appendix.A.4.

### 3.1 OVERVIEW

Given $N$ sparse input images with known camera poses and intrinsics, denoted as $\{(\mathbf{I}_i, \mathbf{E}_i, \mathbf{K}_i)|i = 1, \ldots, N\}$, LVSM synthesizes target image $\mathbf{I}^t$ with novel target camera extrinsics $\mathbf{E}^t$ and intrinsics $\mathbf{K}^t$. Each input image has shape $\mathbb{R}^{H \times W \times 3}$, where $H$ and $W$ are the image height and width (and there are 3 color channels).

**Framework.** As shown in Fig. 2, our LVSM method uses an end-to-end transformer model to directly render the target image. LVSM starts by tokenizing the input images. We first compute a pixel-wise Plücker ray embedding (Plucker, 1865) for each input view using the camera poses and intrinsics. We denote these Plücker ray embeddings as $\{\mathbf{P}_i \in \mathbb{R}^{H \times W \times 6}|i = 1, \ldots, N\}$. We patchify the RGB images and Plücker ray embeddings into non-overlapping patches, following the image tokenization layer of ViT (Dosovitskiy et al., 2020). We denote the image and Plücker ray patches of input image $\mathbf{I}_i$ as $\{\mathbf{I}_{ij} \in \mathbb{R}^{p \times p \times 3}|j = 1, \ldots, HW/p^2\}$ and $\{\mathbf{P}_{ij} \in \mathbb{R}^{p \times p \times 6}|j = 1, \ldots, HW/p^2\}$, respectively, where $p$ is the patch size. For each patch, we concatenate its image patch and Plücker ray embedding patch, reshape them into a 1D vector, and use a linear layer to map it into an input patch token $\mathbf{x}_{ij}$:

$$\mathbf{x}_{ij} = \text{Linear}_{input}([\mathbf{I}_{ij}, \mathbf{P}_{ij}]) \in \mathbb{R}^d, \tag{1}$$

where $d$ is the latent size, and $[\cdot, \cdot]$ means concatenation.

Similarly, LVSM represents the target pose to be synthesized as its Plücker ray embeddings $\mathbf{P}^t \in \mathbb{R}^{H \times W \times 6}$, computed from the given target extrinsics $\mathbf{E}^t$ and intrinsics $\mathbf{K}^t$. We use the same patchify method and another linear layer to map it to the Plücker ray tokens of the target view, denoted as:

$$\mathbf{q}_j = \text{Linear}_{target}(\mathbf{P}^t_j) \in \mathbb{R}^d, \tag{2}$$

where $\mathbf{P}^t_j$ is the Plücker ray embedding of the $j^{\text{th}}$ patch in the target view.

We flatten the input tokens into a 1D token sequence, denoted as $x_1, \ldots, x_{l_x}$, where $l_x = NHW/p^2$ is the sequence length of the input image tokens. We also flatten the target query tokens as $q_1, \ldots, q_{l_q}$ from the ray embeddings, with $l_q = HW/p^2$ as the sequence length.

LVSM then synthesizes a novel view by conditioning on the input view tokens using a full transformer model $M$:

$$y_1, \ldots, y_{l_q} = M(q_1, \ldots, q_{l_q}|x_1, \ldots, x_{l_x}). \tag{3}$$

Specifically, the output token $y_j$ is an updated version of $q_j$, containing information for predicting the pixel values of the $j$th patch of the target view. More details of model $M$ are described in Sec. 3.2.

We recover the spatial structure of output tokens using the inverse operation of the flatten operation. To regress RGB values of the target patch, we employ a linear layer followed by a sigmoid function:

$$\hat{\mathbf{I}}_j^t = \text{Sigmoid}(\text{Linear}_{out}(y_j)) \in \mathbb{R}^{3p^2}. \tag{4}$$

We reshape the predicted RGB values back to the 2D patch in $\mathbb{R}^{p \times p \times 3}$, and then form the synthesized novel view $\hat{\mathbf{I}}^t$ by performing the same operation on all target patches independently.

**Loss Function.**  Following prior works (Zhang et al., 2024; Hong et al., 2024), we train LVSM with photometric novel view rendering losses:

$$\mathcal{L} = \text{MSE}(\hat{\mathbf{I}}^t, \mathbf{I}^t) + \lambda \cdot \text{Perceptual}(\hat{\mathbf{I}}^t, \mathbf{I}^t), \tag{5}$$

where $\lambda$ is the weight for balancing the perceptual loss (Johnson et al., 2016).

## 3.2 TRANSFORMER-BASED MODEL ARCHITECTURE

In this subsection, we present two LVSM architectures—*encoder-decoder* and *decoder-only*—both designed to minimize 3D inductive biases. Following their name, '*encoder-decoder*' first converts input images to an intermediate latent representation before decoding the final image pixels, whereas '*decoder-only*' directly outputs the synthesized target view without an intermediate representation, further minimizing inductive bias in its design. Unlike most standard language model transformers (Vaswani et al., 2017; Jaegle et al., 2021; Radford et al., 2019), which typically use full attention for encoders and causal/cross attention for decoders, we adopt dense **full self-attention** across all our encoder and decoder architectures. The naming of our models is based on their output characteristics, instead of being strictly tied to the transformer architecture they utilize. Please refer to Appendix A.1 for a further detailed discussion of the naming.

**Encoder-Decoder Architecture.**  The encoder-decoder LVSM comes with a learned latent scene representation for view synthesis, avoiding the use of NeRF, 3DGS, and other representations. The encoder first maps the input tokens to an intermediate 1D array of latent tokens (serving as a latent scene representation). The decoder then predicts the outputs, conditioning on the latent tokens and target pose.

Similar to the triplane tokens in LRMs (Hong et al., 2024; Wei et al., 2024), we use $l$ learnable latent tokens $\{e_k \in \mathbb{R}^d | k = 1, ..., l\}$ to aggregate information from input tokens $\{x_i\}$. The encoder, denoted as $\text{Transformer}_{Enc}$, uses multiple transformer layers with self-attention. We concatenate $\{x_i\}$ and $\{e_k\}$ as the input to $\text{Transformer}_{Enc}$, which performs information aggregation between them to update $\{e_k\}$. The output tokens that correspond to the latent tokens, denoted as $\{z_k\}$, are used as the intermediate latent scene representation. The other tokens (updated from $\{x_i\}$, denoted as $\{x_i'\}$) are unused and discarded.

The decoder uses multiple transformer layers with self-attention. In detail, the inputs are the concatenation of the latent tokens $\{z_k\}$ and the target view query tokens $\{q_j\}$. By applying self-attention transformer layers over the input tokens, we get output tokens with the same sequence length as the input. The output tokens that corresponds to the target tokens $q_1, \ldots, q_{lq}$ are treated as final outputs $y_1, \ldots, y_{lq}$, and the other tokens (updated from $\{z_i\}$, denoted as $\{z_i'\}$) are unused. This architecture can be formulated as:

$$x_1', \ldots, x_{l_x}', z_1, \ldots, z_l = \text{Transformer}_{Enc}(x_1, \ldots, x_{l_x}, e_1, \ldots, e_l) \tag{6}$$

$$z_1', \ldots, z_l', y_1, \ldots, y_{l_q} = \text{Transformer}_{Dec}(z_1, \ldots, z_l, q_1, \ldots, q_{l_q}). \tag{7}$$

**Decoder-Only Architecture.**  Our alternate, decoder-only model further eliminates the need for an intermediate scene representation. Its architecture is similar to the decoder in the encoder-decoder architecture but differs in inputs and model size. We concatenate the two sequences of input tokens $\{x_i\}$ and target tokens $\{q_j\}$. The final output $\{y_j\}$ is the decoder's corresponding output for the target tokens $\{q_j\}$. The other tokens (updated from $\{x_i\}$, denoted as $\{x_i'\}$) are unused and discarded. This architecture can be formulated as:

$$x_1', \ldots, x_{l_x}', y_1, \ldots, y_{l_q} = \text{Transformer}_{Dec\text{-}only}(x_1, \ldots, x_{l_x}, q_1, \ldots, q_{l_q}) \tag{8}$$

Here $\text{Transformer}_{Dec\text{-}only}$ has multiple full self-attention transformer layers.

Table 1: **Quantitative comparisons on object-level (left) and scene-level (right) view synthesis.** For the object-level comparison, we matched the baseline settings with GS-LRM (Zhang et al., 2024) in both input and rendering under both resolution of 256 (Res-256) and 512 (Res-512). For the scene-level comparison, we use the same validation dataset used by pixelSplat (Charatan et al., 2024), which has 256 resolution.

| | ABO (Collins et al., 2022a) | | | GSO (Downs et al., 2022) | | | | RealEstate10k (Zhou et al., 2018) | | |
| --- | --- | --- | --- | --- | --- | --- | --- | --- | --- | --- |
| | PSNR ↑ | SSIM ↑ | LPIPS ↓ | PSNR ↑ | SSIM ↑ | LPIPS ↓ | | PSNR ↑ | SSIM ↑ | LPIPS ↓ |
| Triplane-LRM (Li et al., 2023) (Res-512) | 27.50 | 0.896 | 0.093 | 26.54 | 0.893 | 0.064 | pixelNeRF (Yu et al., 2021) | 20.43 | 0.589 | 0.550 |
| GS-LRM (Zhang et al., 2024) (Res-512) | 29.09 | 0.925 | 0.085 | 30.52 | 0.952 | 0.050 | GPNR (Suhail et al., 2022a) | 24.11 | 0.793 | 0.255 |
| Ours Encoder-Decoder (Res-512) | 29.81 | 0.913 | 0.065 | 29.32 | 0.933 | 0.052 | Du et. al (Du et al., 2023) | 24.78 | 0.820 | 0.213 |
| Ours Decoder-Only (Res-512) | 32.10 | 0.938 | 0.045 | 32.36 | 0.962 | 0.028 | pixelSplat (Charatan et al., 2024) | 26.09 | 0.863 | 0.136 |
| LGM (Tang et al., 2024) (Res-256) | 20.79 | 0.813 | 0.158 | 21.44 | 0.832 | 0.122 | MVSplat (Chen et al., 2024) | 26.39 | 0.869 | 0.128 |
| GS-LRM (Zhang et al., 2024) (Res-256) | 28.98 | 0.926 | 0.074 | 29.59 | 0.944 | 0.051 | GS-LRM (Zhang et al., 2024) | 28.10 | 0.892 | 0.114 |
| Ours Encoder-Decoder (Res-256) | 30.35 | 0.923 | 0.052 | 29.19 | 0.932 | 0.046 | Ours Encoder-Decoder | 28.58 | 0.893 | 0.114 |
| Ours Decoder-Only (Res-256) | 32.47 | 0.944 | 0.037 | 31.71 | 0.957 | 0.027 | Ours Decoder-Only | 29.67 | 0.906 | 0.098 |

# 4 EXPERIMENTS

We first describe the datasets we use and baseline methods we compare to, then present the results of LVSM for both object-level and scene-level novel view synthesis.

## 4.1 DATASETS

We train (and evaluate) LVSM on object-level and scene-level datasets separately.

**Object-level Datasets.** We use the Objaverse dataset (Deitke et al., 2023) to train LVSM. We follow the rendering settings in GS-LRM (Zhang et al., 2024) and render 32 random views of 730K objects. We test on two object-level datasets, Google Scanned Objects (Downs et al., 2022) (GSO) and Amazon Berkeley Objects (Collins et al., 2022b) (ABO). GSO and ABO contain 1,099 and 1,000 objects, respectively. Following Instant3D (Li et al., 2023) and GS-LRM (Zhang et al., 2024), we use 4 sparse views as test inputs and another 10 views as target images.

**Scene-level Datasets.** We use the RealEstate10K dataset (Zhou et al., 2018), which contains 80K video clips curated from 10K Youtube videos of both indoor and outdoor scenes. We follow the train/test data split used in pixelSplat (Charatan et al., 2024).

## 4.2 TRAINING DETAILS

**Improving Training Stability.** We observe that LVSM training crashes with plain transformer layers (Vaswani et al., 2017) due to exploding gradients. We empirically find that using QK-Norm (Henry et al., 2020) in the transformer layers stabilizes training. This observation is consistent with Bruce et al. (2024) and Esser et al. (2024). We also skip optimization steps with gradient norm > 5.0 in addition to the standard 1.0 gradient clipping.

**Efficient Training Techniques.** We use FlashAttention-v2 (Dao, 2023) in the xFormers (Lefaudeux et al., 2022), gradient checkpointing (Chen et al., 2016), and mixed-precision training with Bfloat16 data type to accelerate training.

**Other Details.** For more model and training details, please refer to Appendix A.2, and a detailed model architecture diagram (Fig. 8).

## 4.3 COMPARISON TO BASELINES

In this section, we describe our experimental setup and datasets (Sec. 4.1), introduce our model training details (Sec. 4.2), report evaluation results (Sec. 4.3) and perform an ablation study (Sec. 4.4).

**Object-Level Results.** We compare with Instant3D's Triplane-LRM (Li et al., 2023) and GS-LRM (Zhang et al., 2024) at a resolution of 512. As shown on the left side of Table 1, our LVSM method achieves the best performance. In particular, at 512 resolution, our decoder-only LVSM achieves a 3 dB and 2.8 dB PSNR gain against the best prior method GS-LRM on ABO and GSO, respectively; our encoder-decoder LVSM achieves performance comparable to GS-LRM.

We also compare with LGM (Tang et al., 2024) at a resolution of 256, as the official LGM model is trained with that resolution. We also report the performance of models trained on the resolution of 256. Compared with the best prior work GS-LRM, our decoder-only LVSM demonstrates a 3.5 dB and 2.2 dB PSNR gain on ABO and GSO, respectively; our encoder-decoder LVSM yields slightly better performance than GS-LRM.

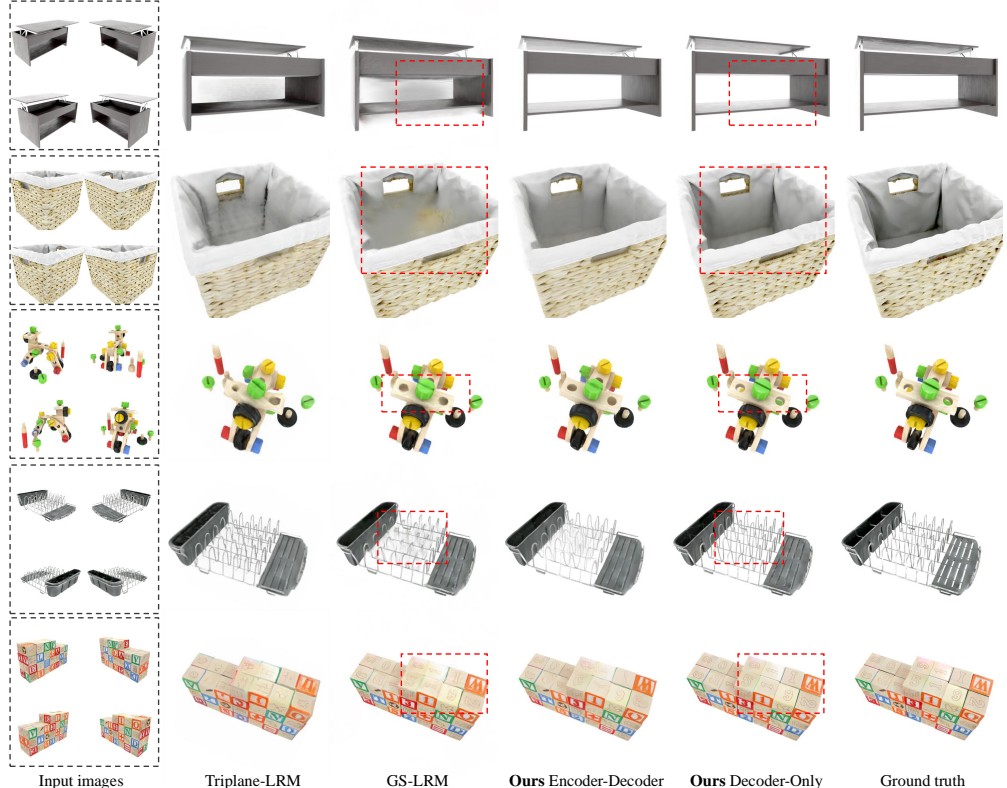

Figure 3: **Object-level visual comparison at 512 resolution.** Given 4 sparse input posed images (leftmost column), we compare our high-res object-level novel-view rendering results with two baselines: Instant3D's *Triplane-LRM* (Li et al., 2023) and *GS-LRM* (Res-512) (Zhang et al., 2024). Both our Encoder-Decoder and Decoder-Only models exhibit fewer floaters (first example) and fewer blurry artifacts (second example), compared to the baselines. Our Decoder-Only model effectively handles complex geometry, including small holes (third example) and thin structures (fourth example). Additionally, it preserves high-frequency texture details (last example).

These significant performance gains validate the effectiveness of our design target of removing 3D inductive bias. More interestingly, the larger performance gain on ABO suggests that LVSM can handle challenging materials, which are difficult for current handcrafted 3D representations. The qualitative results in Fig. 3 and Fig. 7 also validate the high degree of realism of LVSM, especially for examples with specular materials, detailed textures, and thin, complex geometry.

**Scene-Level Results.** We compare on scene-level inputs with pixelNeRF (Yu et al., 2021), GPNR (Suhail et al., 2022a), Du et al. (2023), pixelSplat (Charatan et al., 2024), MVSplat (Chen et al., 2024) and GS-LRM (Zhang et al., 2024). As shown on the right side of Table 1, our decoder-only LVSM shows a 1.6 dB PSNR gain compared with the best prior work, GS-LRM. Our encoder-decoder LVSM also demonstrates comparable results to GS-LRM. These improvements can be observed qualitatively in Fig. 4, where LVSM has fewer floaters and better performance on thin structures and specular materials, consistent with the object-level results. These outcomes again validate the efficacy of our design of using minimal 3D inductive bias.

**LVSM Trained with Only 1 GPU.** Limited computing is a key bottleneck for academic research. To show the potential of LVSM using academic-level resources, we train LVSM on the scene-level dataset (Zhou et al., 2018) following the setting of pixelSplat (Charatan et al., 2024) and MVSplat (Chen et al., 2024), with only a single A100 80G GPU for 7 days. In this experiment, we use a smaller decoder-only model (denoted LVSM-small) with 6 transformer layers and a smaller batch size of 64 (in contrast to the default setting of 24 layers and batch size 512). Our decoder-only LVSM-small shows a performance of 27.66 dB PSNR, 0.870 SSIM, and 0.129 LPIPS. This performance surpasses the prior best 1-GPU-trained model, MVSplat, with a 1.3 dB PSNR gain. We also train a decoder-only LVSM (12 transformer layers, batch size 64) with 2 GPUs for 7 days, exhibiting a performance of 28.56 dB PSNR, 0.889 SSIM, and 0.112 LPIPS. This performance is

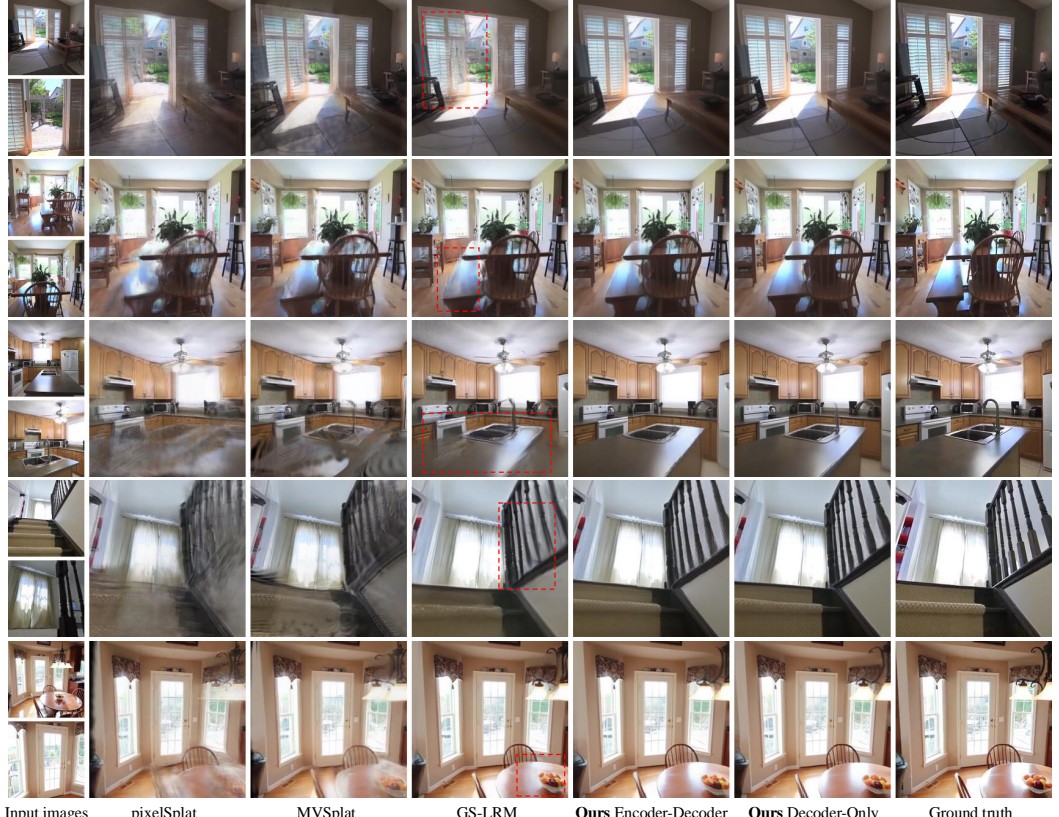

Input images    pixelSplat    MVSplat    GS-LRM    **Ours** Encoder-Decoder    **Ours** Decoder-Only    Ground truth

Figure 4: **Scene-level visual comparison.** We evaluate encoder-decoder and decoder-only LVSM on scene-level view synthesis, comparing them to the prior leading baseline methods, namely pixelSplat (Charatan et al., 2024), MVSplat (Chen et al., 2024), and GS-LRM (Zhang et al., 2024). Our results exhibit fewer texture and geometric artifacts, feature more realistic specular reflections, and are closer to the ground truth images.

even better than GS-LRM with 24 transformer layers trained on 64 GPUs. These results show the promising potential of LVSM for academic research.

### 4.4 ABLATION STUDIES

**Model Size.** In Tab. 2, we ablate the model size designs of both LVSM variants on both object and scene level. To save resources, these experiments are run with 8 GPUs and a total batch size of 64.

For the encoder-decoder LVSM, we maintain the total number of transformer layers while allocating a different number of layers to the encoder and decoder. We observe that using more decoder layers helps performance while using more encoder layers harms performance. We hypothesize that this is because the encoder uses the latent representation to compress the input image information, and a deeper encoder makes this compression process harder to learn, resulting in greater compression errors. This observation suggests that using the inductive bias of the encoder and intermediate latent representation may not be optimal for the final quality, aligning with our observation that the decoder-only variant outperforms the encoder-decoder variant.

For the decoder-only LVSM, we experiment with using different numbers of transformer layers and model sizes in the decoder. The experiment verifies that the decoder-only LVSM shows increasing performance when using more transformer layers, and validates the scalability of the decoder-only LVSM.

**Model Architecture.** As shown in Tab. 3, we evaluate the effectiveness of our model designs. We also visualize the equivalent attention mask for each design in Fig. 9 for better illustration. To save resources, the encoder-decoder experiments here are run with 32 GPUs, a total batch size of 256, and a decreased number of training target views of 8. The decoder-only experiment is run with our original setup.

Table 2: **Ablations studies on model sizes.** The following experiments are all run with 8 GPUs and a total batch size of 64.

| | # Params | GSO PSNR ↑ | SSIM ↑ | LPIPS ↓ | RealEstate10k PSNR ↑ | SSIM ↑ | LPIPS ↓ |
|---|---|---|---|---|---|---|---|
| Ours Encoder-Decoder (6 + 18) | 173M | 26.48 | 0.901 | 0.065 | 28.32 | 0.888 | 0.117 |
| Ours Encoder-Decoder (12 + 12) | 173M | 25.69 | 0.889 | 0.076 | 27.39 | 0.869 | 0.137 |
| Ours Encoder-Decoder (18 + 6) | 173M | 24.74 | 0.873 | 0.091 | 26.80 | 0.855 | 0.152 |
| Ours Decoder-Only (24 layers) | 171M | 27.04 | 0.910 | 0.055 | 28.89 | 0.894 | 0.108 |
| Ours Decoder-Only (18 layers) | 128M | 26.81 | 0.907 | 0.057 | 28.77 | 0.892 | 0.109 |
| Ours Decoder-Only (12 layers) | 86M | 26.11 | 0.896 | 0.065 | 28.61 | 0.890 | 0.111 |
| Ours Decoder-Only (6 layers) | 43M | 24.15 | 0.865 | 0.092 | 27.62 | 0.869 | 0.129 |

Table 3: **Ablations studies on model architecture.**

| | GSO (Downs et al., 2022) PSNR ↑ | SSIM ↑ | LPIPS ↓ |
|---|---|---|---|
| Ours Encoder-Decoder | **28.07** | **0.920** | **0.053** |
| Ours w/ CNN tokenizer | 27.59 | 0.914 | 0.052 |
| Ours w/o latents' updating | 26.61 | 0.903 | 0.061 |
| Ours w/ per-patch prediction | 26.27 | 0.897 | 0.072 |
| Ours w/ pure cross-att decoder | 24.60 | 0.876 | 0.085 |

| | RealEstate10k (Zhou et al., 2018) PSNR ↑ | SSIM ↑ | LPIPS ↓ |
|---|---|---|---|
| Ours Decoder-Only | **29.67** | **0.906** | **0.098** |
| Ours w/ per-patch prediction | 28.98 | 0.897 | 0.103 |

Our encoder-decoder LVSM leverages an encoder to transform input images into a set of 1D tokens that serve as an intermediate latent representation of the 3D scene. A decoder can then render novel view images from this latent representation. While the encoder-decoder LVSM shares high-level conceptual similarities with SRT (Sajjadi et al., 2022)—both are based on transformers and use 1D latent tokens as an intermediate latent representation without an explicit 3D representation—our encoder-decoder LVSM introduces a very distinct architecture that significantly enhances performance. In the following paragraphs, we evaluate many of our design decisions and also consider some alternate components based on SRT, showing that our design yields significant improvements.

**Tokenization Strategy for Encoder Input:** To generate input tokens for the encoder, we draw inspiration from ViT (Dosovitskiy et al., 2020) and recent LRMs (Wei et al., 2024; Zhang et al., 2024). Specifically, we tokenize the input images and their poses by simply splitting the concatenated input views and Plücker ray embeddings into non-overlapping patches. In contrast, SRT relies on shallow convolutional neural networks (CNNs) to extract patch features, which are then flattened into tokens. As an ablation study, we tried SRT's CNN-based tokenizing method and observed that it makes training more unstable with a larger grad norm, which leads to worse performance. As demonstrated in Tab. 3, replacing our simple tokenizer with SRT's CNN-based tokenizer degrades performance (ours w/ CNN tokenizer).

**Fixed-Length Latent Encoding for Efficient Rendering:** Our encoder employs self-attention to progressively compress the information from posed input images into a fixed-length set of 1D latent tokens. This design ensures a consistent rendering speed, regardless of the number of input images, as shown in Fig. 6. This differs from approaches like SRT where latent token size increases linearly with input views, reducing rendering efficiency as the number of views grows.

**Bidirectional Self-Attention Decoder:** The decoder of our encoder-decoder model utilizes pure (bidirectional) self-attention, allowing latent tokens and output target image tokens to attend to each other. This enables i) latent tokens to be updated by fusing information from themselves and from other tokens, which also means the parameters of the decoder are a part of the scene representation; ii) output patch pixels can also attend to other patches for joint updates, ensuring the global awareness of the rendered target image. We ablate our full-attention design choice by experimenting with different attention mechanisms, illustrated in Fig. 9. As shown in Tab. 3, disabling either the latents' updating (ours w/o latents' updating) or the joint updating of direct output pixel patches (ours w/ per-patch prediction) significantly degrades performance. SRT cannot support either mechanism because it employs a decoder with pure cross-attention. We experiment with an LVSM variant by adopting SRT's decoder designs. As shown in Tab. 3 (ours w/ pure cross-att decoder), this modification leads to reduced performance.

The decoder-only LVSM further pushes towards eliminating inductive bias, bypassing an intermediate representation altogether. It adopts a single-stream transformer to directly convert the input multi-view tokens into target view tokens, treating view synthesis like a sequence-to-sequence translation task, which is fundamentally different from prior work. We ablate the importance of the joint prediction of target image patches in the decoder-only LVSM. We design a variant where the colors of each target pose patch are predicted independently, without applying self-attention across other target pose patches. We achieve this by letting each transformer layer's key and value matrices only consist of updated input image tokens, while both the updated input image tokens and target pose tokens form the query matrices. As shown on the bottom part of Tab. 3, this variant shows worse performance, with a 0.7 dB PSNR degradation. This result demonstrates the importance of using both input and target tokens as context tokens for information exchange using the simplest full self-attention transformer, which is consistent with our philosophy of reducing inductive bias.

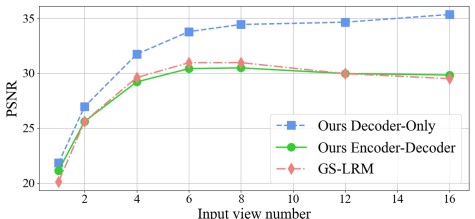
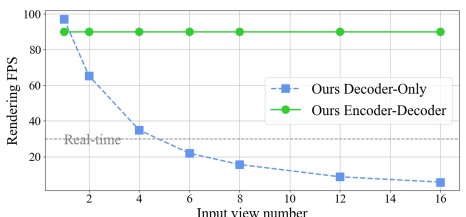

Figure 5: **Zero-shot generalization to different number of input images** on the GSO dataset (Downs et al., 2022). We note that all models are trained with just 4 input views.

Figure 6: **Rendering FPS with different number of input images.** We test the FPS on the object level under $256 \times 256$ resolution. We refer to rendering as the decoding process, which synthesizes novel views from latent tokens or input images.

## 4.5 DISCUSSION

**Zero-shot Generalization to More Input Views.** We compare our LVSM with GS-LRM by taking different numbers of input views during inference. We report the results on object-level inputs. Note that these models are trained with 4 input views, and tested on varying numbers of input views in a zero-shot manner. As shown in Fig. 5, our decoder-only LVSM shows increasingly better performance when using more input views, verifying the scalability of our model design at test time. Our encoder-decoder LVSM shows a similar pattern as GS-LRM, i.e., it exhibits a performance drop when using more than 8 input views. We conjecture that this is due to the inductive bias of the encoder-decoder design, i.e., using intermediate representation as a compression of input information limits performance. In addition, our single-input results (input view number = 1) are competitive and even outperform some of the baseline that takes 4 images as input. These performance patterns validate our design target of using minimal 3D inductive bias for learning a fully data-driven rendering model and cohere with our discussion in Sec. 3.2.

**Encoder-Decoder versus Decoder-Only.** As mentioned in Sec. 3, the decoder-only and encoder-decoder architectures exhibit different trade-offs in speed, quality, and potential.

The encoder-decoder model transforms 2D image inputs into a fixed-length set of 1D latent tokens, which serve as a compressed representation of the 3D scene. This approach simplifies the decoder, reducing its model size. Furthermore, during the rendering/decoding process, the decoder always receives a fixed number of tokens, regardless of the number of input images, ensuring a consistent rendering speed. As a result, this model offers improved rendering efficiency, as shown in Fig. 6. Additionally, the use of 1D latent tokens as the latent representation for the 3D scene opens up the possibility of integrating this model with generative approaches for 3D content generation on its 1D latent space. Nonetheless, the compression process can result in information loss, as the fixed latent token length is usually smaller than the original image token length, imposing an upper bound on performance. This characteristic of the encoder-decoder LVSM mirrors prior encoder-decoder LRMs; however, unlike those LRMs, our encoder-decoder LVSM does not have an explicit 3D structure.

In contrast, the decoder-only model learns a direct mapping from the input image to the target novel view, showcasing better scalability. For example, as the number of input images increases, the model can leverage all available information, resulting in improved novel view synthesis quality. However, this property also leads to a linear increase in input image tokens, leading to quadratic growth in computational complexity and limiting rendering speed.

**Single Input Image.** As shown in our project page, Fig. 1 and Fig. 5, we observe that our LVSM also works with a single input view for many cases, even though the model is trained with multi-view images during training. This observation shows the capability of LVSM to understand the 3D world, e.g., understanding depth, rather than performing purely pixel-level view interpolation.

## 5 CONCLUSION

In this work, we presented the Large View Synthesis Model (LVSM), a transformer-based approach designed to minimize 3D inductive biases for scalable and generalizable novel view synthesis. Our two architectures—encoder-decoder and decoder-only—bypass physical-rendering-based 3D representations like NeRF and 3D Gaussian Splatting, allowing the model to learn priors directly from data, leading to more flexible and scalable novel view synthesis. The decoder-only LVSM, with its minimal inductive biases, excels in scalability, zero-shot generalization, and rendering quality, while the encoder-decoder LVSM achieves faster inference due to its fully learned latent scene representation. Both models demonstrate superior performance across diverse benchmarks and mark an important step towards general and scalable novel view synthesis in complex, real-world scenarios.

**Acknowledgements.** We thank Kalyan Sunkavalli for helpful discussions and support. This work was done when Haian Jin, Hanwen Jiang, and Tianyuan Zhang were research interns at Adobe Research. This work was also partly funded by the National Science Foundation (IIS-2211259, IIS-2212084).

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

# A APPENDIX

## A.1 NAMING CLARIFICATION

Importantly, we clarify that the naming of '*encoder*' and '*decoder*' are based on their output characteristics—i.e., the encoder outputs the latent while the decoder outputs the target—rather than being strictly tied to the transformer architecture they utilize. For instance, in the encoder-decoder model, the decoder consists of multiple transformer layers with self-attention (referred to as Transformer Encoder layers in the original transformer paper). However, we designate it as a decoder because its primary function is to output results. These terminologies align with conventions used in LLMs (Vaswani et al., 2017; Radford et al., 2019; Devlin et al., 2019). Notably, we apply self-attention to all tokens in every transformer block of both models without introducing special attention masks or other architectural biases, in line with our philosophy of minimizing inductive bias.

## A.2 ADDITIONAL IMPLEMENTATION DETAILS

We train LVSM with 64 A100 GPUs with a batch size of 8 per GPU. We use a cosine learning rate schedule with a peak learning rate of 4e-4 and a warmup of 2500 iterations. We train LVSM for 80k iterations on the object and 100k on scene data. LVSM uses a image patch size of $p = 8$ and token dimension $d = 768$. The details of the transformer layers follow GS-LRM(Zhang et al., 2024) with an additional QK-Norm. Unless noted, all models have 24 transformer layers, the same as GS-LRM. The *encoder-decoder* LVSM has 12 encoder layers and 12 decoder layers, with 3072 latent tokens. Note that our model size is smaller than GS-LRM, as GS-LRM uses a token dimension of 1024.

For object-level experiments, we use 4 input views and 8 target views for each training example by default. We first train with a resolution of 256, which takes 4 days for the *encoder-decoder* model and 7 days for the *decoder-only* model. Then, we finetune the model with a resolution of 512 for 10k iterations with a smaller learning rate of 4e-5 and a smaller total batch size of 128, which takes 2.5 days. For scene-level experiments We use 2 input views and 6 target views for each training example. We first train with a resolution of 256, which takes about 3 days for both *encoder-decoder* and *decoder-only* models. Then, we finetune the model with a resolution of 512 for 20k iterations with a smaller learning rate of 1e-4 and a total batch size of 128 for 3 days. For both object and scene-level experiments, the view selection details and camera pose normalization methods follow GS-LRM. We use a perceptual loss weight $\lambda$ as 0.5 and 1.0 on scene-level and object-level experiments, respectively.

We do not use bias terms in our model, for both Linear and LayerNorm layers. We initialize the model weights with a normal distribution of zero-mean and standard deviation of $0.02/(2*(idx+1))**0.5$, where *idx* means transform layer index.

We train our model with AdamW optimizer (Kingma, 2014). The $\beta_1$ and $\beta_2$ are set to 0.9 and 0.95 respectively, following GS-LRM. We use a weight decay of 0.05 on all parameters except the weights of LayerNorm layers.

## A.3 ADDITIONAL VISUAL RESULTS

We show the visualization of LVSM at the object level with 256 resolution in Fig. 7. Consistent with the findings of the experiment with 512 resolution (Fig. 3), LVSM performs better than the baselines on texture details, specular material, and concave geometry.

## A.4 DETAILED MODEL ARCHITECTURE

We have provided a detailed model architecture figure, as shown in Fig. 8.

## A.5 ATTENTION MASK ILLUSTRATION FOR DIFFERENT DESIGN CHOICES

In Fig. 9, we visualize the corresponding attention masks for the various design choices discussed in Sec. 4.4.

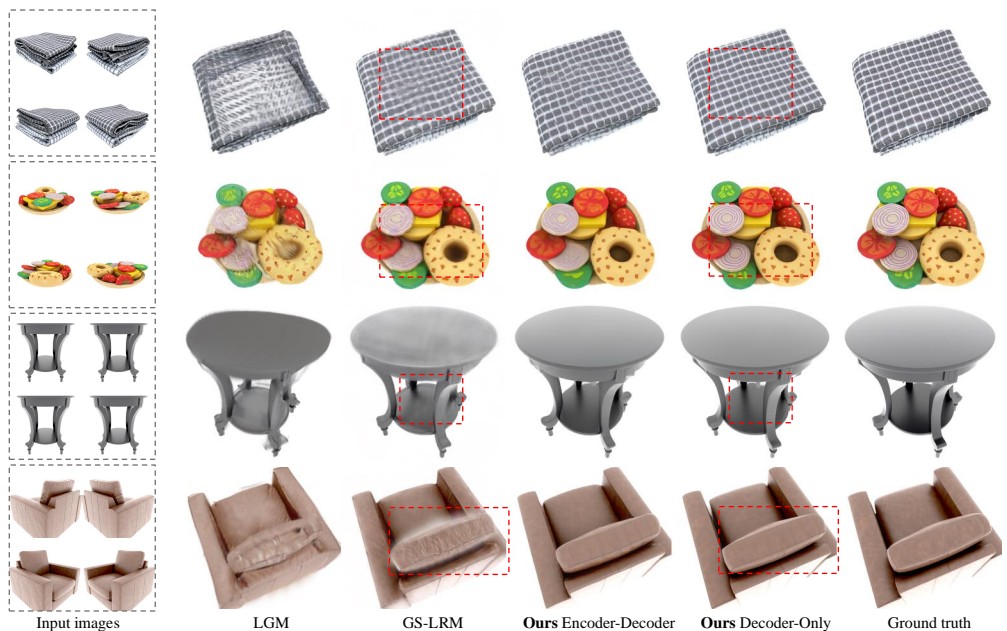

Figure 7: **Object-level visual comparison at 256 resolution.** Comparing with the two baselines: *LGM*(Tang et al., 2024) and *GS-LRM* (Res-256) (Zhang et al., 2024), both our Encoder-Decoder and Decoder-Only models have fewer floater artifacts (last example), and can generate more accurate view-dependent effects (third example). Our Decoder-Only model can better preserve the texture details (first two examples).

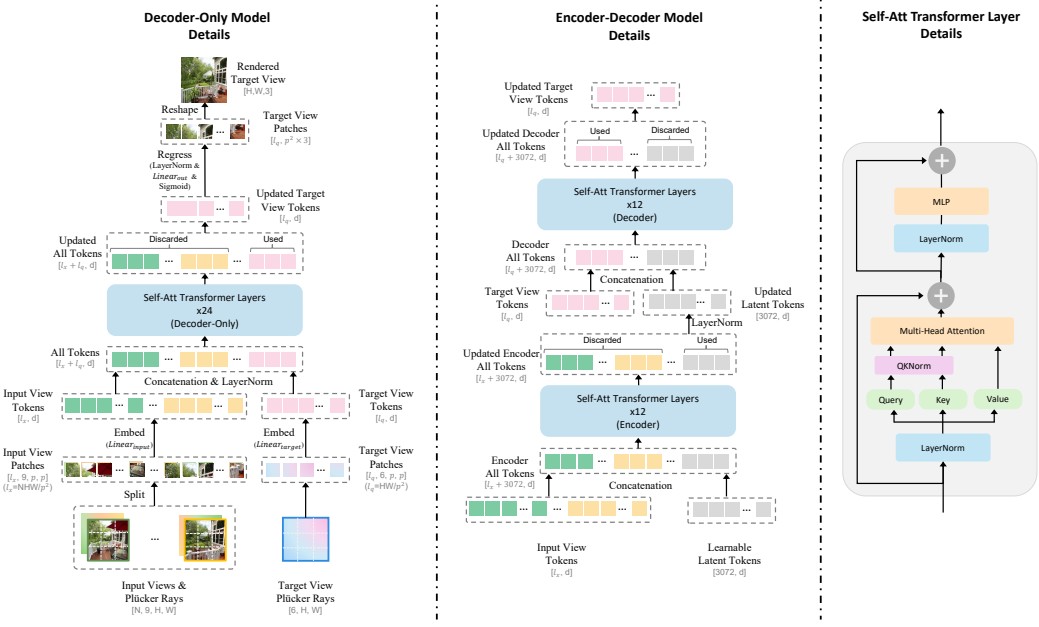

Figure 8: **Model Details.** We introduce two architectures: (1) an encoder-decoder LVSM, which encodes input image tokens into a fixed number of 1D latent tokens, functioning as a fully-learned latent scene representation, and decodes novel-view images from them; and (2) a decoder-only LVSM, which directly maps input images to novel view outputs, completely eliminating intermediate scene representations. Both models consist of pure self-attention blocks.

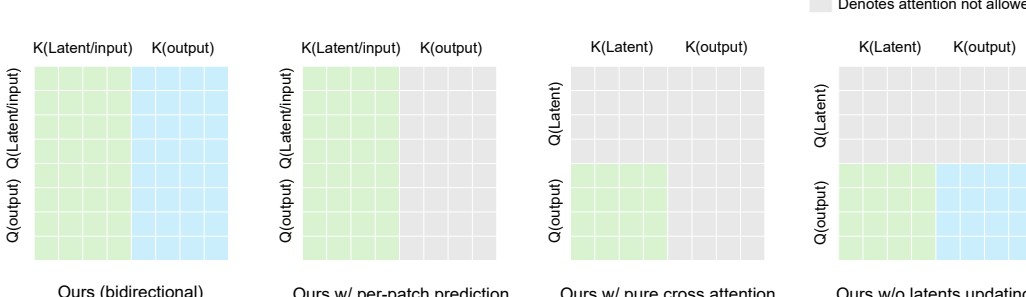

Figure 9: **Attention Mask Visualization.** Both our encoder-decoder and decoder-only models employ bidirectional self-attention modules. This figure visualizes the corresponding attention masks for the various design choices discussed in Sec. 4.4. (We use green color for the columns of latent/input tokens and blue for the columns of output pixel patch tokens.) In our encoder-decoder architecture, the decoder utilizes pure self-attention, enabling latent tokens and different output target image tokens to jointly attend to each other. Consequently, latent tokens can be updated across transformer layers, while different output patch pixels can also attend to each other for joint updates. As shown in Table 3, disabling either the joint updating of output patch pixels (ours w/ per-patch prediction) or the latents' updating (ours w/o latents' updating) significantly degrades performance. Prior work, SRT (Sajjadi et al., 2022), eliminates both mechanisms by employing a decoder with pure cross-attention (ours w/ pure cross-att decoder), leading to even worse performance. Similarly, in our decoder-only model, disabling the joint updating of output patch pixels (ours w/ per-patch prediction) also results in a notable performance drop.

## A.6    DISCUSSION OF DIFFERENCES WITH PRIOR GENERATIVE NVS MODELS

Motivated by the success of the previous NVS geometry-free approaches (Sitzmann et al., 2021; Sajjadi et al., 2022), and the effectiveness of diffusion models in image-to-image tasks (Saharia et al., 2022a; Ramesh et al., 2022; Saharia et al., 2022b), 3DiM (Watson et al., 2022) explores training image-to-image diffusion models for object-level multi-view rendering to perform novel view synthesis without an explicit 3D representation. However, 3DiMs is trained from scratch using limited 3D data (Sitzmann et al., 2019), limiting it to category-specific settings and without zero-shot generalization.

The following work Zero-1-to-3 (Liu et al., 2023a) adopts a similar pipeline without a 3D representation but fine-tunes its model from a pretrained 2D diffusion model using a larger 3D object dataset (Deitke et al., 2023), achieving better generalization and higher quality. However, Zero-1-to-3's view synthesis results suffer from inherent multi-view inconsistency because it is a probabilistic model and it generates one target image at a time independently.

To improve this inconsistency problem, several works (Liu et al., 2023b; Yang et al., 2023; Ye et al., 2023; Tung et al., 2024) integrate additional forms of 3D inductive bias, such as a 3D representation, epipolar attention, etc., into the diffusion denoising process, leading to increased computational cost. Other approaches (Li et al., 2023; Shi et al., 2023a;b; Long et al., 2023) predict a single image grid representing (specific) multi-view images with fixed camera pose, sacrificing the ability to control the camera. More recent works, including Free3D (Zheng & Vedaldi, 2024), EscherNet (Kong et al., 2024), CAT3D (Gao* et al., 2024), SV3D (Voleti et al., 2025), and some other video model based work (Wang et al., 2023b; He et al., 2024; Yu et al., 2024a; Yan et al., 2024), jointly predict multiple target views with accurate camera control while ensuring view consistency by integrating cross-view attention. However, these methods guarantee consistency only for the finite set of jointly predicted views.

In contrast, our generalizable deterministic models do not possess the same inherent inconsistency issues of probabilistic models. As demonstrated in the video results on our project webpage, after being trained on large-scale multi-view data, our models can independently generate each target image with precise camera control while maintaining view consistency—without relying on the cross-view attention mechanisms employed by previous generative models. This capability enables our models to generate an unlimited number of consistent views for the observed regions of reconstructed scenes,

unlike prior generative models. Nonetheless, our deterministic models have their own inherent limitations, i.e., they can't hallucinate unseen regions, which are discussed in Appendix A.7.

## A.7 LIMITATIONS

Our models are deterministic, and like all prior deterministic approaches (Chen et al., 2021; Wang et al., 2021a; Sajjadi et al., 2022; Wang et al., 2023a; Zhang et al., 2024), they struggle to produce high-quality results in unseen regions. Previous 3D-based deterministic models typically generate blurry artifacts for those regions due to uncertainty, whereas our model often generates noisy and flickering artifacts with fixed patterns. To illustrate this, we provide video examples of related failure cases on our webpage. Incorporating generative techniques or combining generative methods with our model could help solve this issue, which we leave as a promising future direction.

Additionally, our model's performance degrades when provided with images with aspect ratios and resolutions different from those seen during training. For instance, when trained on $512 \times 512$ images and tested on $512 \times 960$ input images, we observe high-quality novel view synthesis at the center of the output but blurred regions at the horizontal boundaries that extend beyond the training aspect ratio. We hypothesize that this limitation arises because our model is trained on center-cropped images. Specifically, the Plücker ray density is smaller at the boundaries of the image's longer side, and since our model is not trained on such data, it struggles to generalize. Expanding the training dataset to include more diverse image resolutions and aspect ratios could help address this issue.

