# OpenReview forum: "LVSM: A Large View Synthesis Model with Minimal 3D Inductive Bias"
_ICLR.cc/2025/Conference — ICLR 2025 Oral_

### Official Review · Reviewer_jAgM · 2024-10-28

**Soundness:** 4
**Presentation:** 3
**Contribution:** 3
**Rating:** 8
**Confidence:** 4

**Summary:**

This paper proposes LVSM, a general novel view synthesis (NVS) model for sparse-view inputs. It aims to improve the quality, efficiency, and scalability of NVS by minimizing 3D inductive bias. Different from previous works, LVSM directly utilizes transformer-based backbone to synthesize novel views without intermediate 3D representations and corresponding rendering equations. The author(s) introduce two versions of LVSM. The first model, encoder-decoder LVSM, encodes inputs into a scene representation token and decodes it into novel views. The second model, decoder-only LVSM, further removes the need for intermediate representation by adopting a single-stream transformer. Both models patchify the posed input view and use Plucker ray embeddings to tokenize the target view. Experiments show that LVSM trained on 2-4 input views has generalizability to an unseen number of views, and it outperforms the SOTA GS-LRM by 1.5 to 3.5dB PSNR.

**Strengths:**

1.	The proposed models achieve impressive reconstruction results in terms of PSNR, SSIM, LPIPS, and qualitative evaluation.
2.	The proposed models can be trained with a single A100 80G GPU. In contrast, existing general reconstruction models, like LRM, LGM, and GS-LRM, usually require a large number of computational resources.
3.	The proposed model is generalizable to an unseen number of input views, from single view to more than 10 views.
4.	Ablation studies show the scalability of LVSM, where transformers with more layers generally perform better in terms of the quantitative reconstruction results.

**Weaknesses:**

1.	The qualitative results are not consistent with the quantitative results. In specific, Table 1 shows that the encoder-decoder LVSM achieves similar quantitative results as GS-LRM, and sometimes worse in object-level datasets. However, Figure 3 and Figure 7 substantially outperform GS-LRM. I am worried about potential cherry picks in the qualitative results. It will help evaluation by showing more examples as well as some failure cases of LVSM.
2.	Missing experiments for efficiency comparison. Specifically, the paper claims the higher efficiency of LVSM than previous methods, while there is no comparison between LVSM and other NVS models.

**Questions:**

1.	It is commonly believed that 3D reconstruction improves view consistency for novel view synthesis. By removing 3D inductive bias, how does LVSM ensure consistency across novel views? Is this knowledge simply learned using attention mechanism? To demonstrate view consistency, you may want to provide some 3D-aware metrices for comparison, such as reprojection error or depth estimation, which will help my evaluation a lot.
2.	Could you please provide a more detailed diagram for the LVSM architecture? Specifically, such a diagram will help readers understand the methodology more easily. In the submission, only Figure 2 is available with very brief design.
3.	Refer to weakness.
4. Will you release your codes and pretrained models after possible acceptance?

---

> ### Author Response · Authors · 2024-11-24
>
> Thank you for your insightful comments and valuable suggestions. We have revised our paper based on your feedback. Here are our responses to your comments:
>
> 1. **Questions related to view consistency**
>
>     As demonstrated in the video results on our project webpage, our generalizable deterministic models, trained on large-scale multi-view data, are capable of generating high-quality novel view synthesis results while maintaining strong view consistency. This demonstrates that 3D reconstruction or explicit 3D representations are not strictly necessary for achieving view consistency. View consistency can be learned by transformer architecture. The advancements of recent video diffusion can also support this observation.
>
>     For the evaluation of novel view synthesis, we have shown PSNR metrics for the reprojected novel view images for both object and scene, as shown in Table 1. PSNR measures the pixel-level error between our prediction and the ground-truth images, which can work as a good evaluation for the 3D consistency, as done in prior NeRF/3DGS methods.
> 2. **Detailed Diagram**
>
>     We sincerely thank you for your suggestion. We have revised our paper and included a detailed model architecture diagram in Figure 8 of the Appendix. Hope this can help readers better understand our models. Please let us know if something is still unclear.
>
> 3. **Questions regarding the qualitative results are not consistent with the quantitative results.**
>
>     Although our encoder-decoder model and GS-LRM achieve comparable PSNR results, our encoder-decoder model clearly has better perception-based  LPIPS metrics performance, which aligns better with visual quality than PSNR. If the reviewer remains concerned about this issue, we are happy to include additional qualitative results in a future revision of the paper.
>
> 4. **Adding failure cases**
>
>     We appreciate the reviewer's suggestion regarding failure cases. In response, we have added a dedicated limitations section in Appendix A7 and included related failure cases on our anonymous project webpage.
>
> 5. **Efficiency comparison**
>
>      We do not claim that our methods are more efficient than previous approaches. We sincerely apologize for any misunderstanding this may have caused. If the reviewer can point out the specific sections in question, we would be happy to revise the corresponding descriptions.
>
> 6. **Question related to code release**
>
>     We will release our code in the near future.

---

> ### Comment · Reviewer_jAgM · 2024-11-25
>
> Thanks for the author(s)' response. Here are some further comments:
>
> 1. The concern has been well addressed. I believe that the view consistency is properly learned in the transformer after seeing the examples with unseen regions, where the proposed LVSM is consistent at seen regions but suffers from minor jittering at the unseen parts.
> 2. The diagram is clear and easy-to-understand now.
> 3. The claim is reasonable and accepted, while it's better to point out what kind of examples results in better LPIPS but worse PSNR.
> 4. The concern has been well addressed.
> 5. Generalizable models are proposed to increase the efficiency of novel view synthesis and 3D reconstruction. In the 075 line, the author(s) also mentions how they contribute to a scalable and efficient novel view synthesis model. However, I cannot see an obvious motivation about removing 3D inductive bias, if the model increases much computational complexity with comparison to other reconstruction-based large models, e.g. LRM, GS-LRM. Specifically, I'm asking what the largest advantage of removing 3D inductive bias is. This includes but is not limited to:
>     - Is LVSM easier to be trained than GS-LRM? (more stable or faster)
>     - Does LVSM supports faster rendering than other LRMs?
> If the efficiency is not you major claim, it's better to fix the over-claim in 075 line. Also, how much speed sacrifice is needed, compared to reconstruction-based generalizable models? Does LVSM still support real-time rendering?
> 6. Thanks.
>
> I may temporarily lower my score before the (5.) comment is properly addressed.

---

> ### Author Response · Authors · 2024-11-25
>
> Thank you for your additional comments.
>
> We are genuinely pleased to see that our responses have addressed most of your concerns, except for point 5. Here, we provide further clarification:
>
> * *Training Stability and Efficiency*: We found that LVSM is easier to train compared to GS-LRM, with training being more stable and faster. Specifically, during training, the final LVSM with QK-norm generally exhibited a lower gradient norm than GS-LRM with QK-norm, making it much less likely to crash. As said in GS-LRM's paper on page 21, it needs to design a GS initialization carefully with some magic number ***to help training stability***, while LVSM doesn't need to. Additionally, as noted in our Lines 370-410, LVSM trained on 2 GPUs outperformed GS-LRM trained on 64 GPUs, further demonstrating its training efficiency.
>
> * *Rendering Efficiency*: We want to emphasize that we do not intend to claim our model has better rendering efficiency compared to GS-LRM. As shown in Figure 6, we reported our rendering FPS, and we acknowledge that GS-LRM benefits from the explicit GS representation, resulting in better rendering efficiency.
>
> To clarify this, we have revised the term "efficient" in Line 75 to "training-efficient.", and we will further clarify this in the final revised paper.
>
> We sincerely apologize for any misunderstanding this may have caused. As the discussion period deadline approaches, please don't hesitate to share any further concerns. We hope that our responses resolve all your concerns, and we kindly hope you can restore the rating.

---

> ### Comment · Reviewer_jAgM · 2024-11-26
>
> Thanks for the response. The concern in comment (5.) has been resolved.

---

### Official Review · Reviewer_Lz3H · 2024-10-31

**Soundness:** 4
**Presentation:** 3
**Contribution:** 3
**Rating:** 8
**Confidence:** 4

**Summary:**

This paper proposes a novel transformer-based approach for scalable and generalizable novel view synthesis from sparse-view inputs. Two models are proposed. both of which bypass the 3D inductive biases used in previous methods and address novel view synthesis with a fully data-driven approach. Comprehensive evaluations across multiple datasets demonstrate that the work achieve state-of-the-art novel view synthesis quality.

**Strengths:**

1. this paper minimizes 3D inductive biases for scalable and generalizable novel view synthesis;
2. Two LVSM architectures—encoder-decoder and decoder-only—are designed to minimize 3D inductive biases;
3. Both of the two models achieve impressive sparse novel view synthesis performance.
4. The impressive result might provide a potential new representation for generalizable NVS task.

**Weaknesses:**

The architecture level is not well explained and might cause a bit confusing, I will elaborate it in questions part.

**Questions:**

The architecture of decoder model is elaborated from L303-308, the attention layer in the transformer block seems to be self-attention. During both training and inference stage, I wonder how many input and target views are used for each batch sample, will the self-attention between target views affect the result? If I want to get the target view at P1 and P2, will inferencing them at the same feedforward pass be different from inferencing one by one?

---

> ### Author Response · Authors · 2024-11-24
>
> Thank you for your insightful comments and valuable suggestions. We have revised our paper based on your feedback. Here are our responses to your comments:
> 1. **More illustrations for the architecture**
>
>     Thank you for your suggestion. We have now included a detailed model architecture diagram in Figure 8 of the Appendix.
> 2. **Clarification about the decoder architecture**
>
>    (Note: Due to paper edits during rebuttal, the original “L303–308” is now “L262–269”)
> 	(a). Yes, we use the same self-attention block for all encoder-decoder and decoder-only layers.
> 	(b). As said in L889 and L893, for the scene-level experiment, we use 2 input views and 6 target views for each training scene. For the object-level experiment, we use 4 input views and 8 target views for each training object. During both training and inference, our model predicts each target image one at a time. We also experimented with predicting multiple target views simultaneously in a single feedforward pass with our pre-trained model. However, this led to degraded performance. For instance, on the GSO (res 256) dataset, the decoder-only model’s performance dropped as follows:
>
>     * PSNR: From 31.71 to 29.61
>     * LPIPS: From 0.027 to 0.039
>     * SSIM: From 0.957 to 0.943
>
>     This degradation is understandable, as the model is trained to predict only one target image per feed-forward pass, and deviating from this setup during inference affects its performance.

---

> > ### Comment · Reviewer_Lz3H · 2024-11-26
> >
> > Thank you for your additional comments. My questions are solved. I will keep the original rating.

---

> > > ### Author Response · Authors · 2024-11-28
> > >
> > > Thank you for taking the time to review our responses and confirm that your questions have been resolved. We appreciate your thoughtful feedback and support!

---

### Official Review · Reviewer_NhwW · 2024-10-31

**Soundness:** 3
**Presentation:** 3
**Contribution:** 2
**Rating:** 8
**Confidence:** 4

**Summary:**

The authors propose to train a generalizable novel view synthesis model with minimal 3D inductive biases, training on posed multi-view data to map images and camera rays to novel views, with target rays as queries.  This is supervised by a simple photometric and perceptual novel view loss.
The architecture uses ViT-like image tokenization to embed images and corresponding Plucker rays, and the target views are decoded by Plucker ray queries.  Two Transformer architectures are considered – decoder-only, and encoder-decoder.
These are compared to prior work, and differences in performance between the two choices are compared and contrasted.

**Strengths:**

The model variants introduced compare well to prior work; the large improvements compared to baselines for the decoder-only variant are impressive.  The investigation of both encoder-decoder and decoder-only as modeling choices (with the discussed tradeoffs) is also interesting and seems novel.  The discussion of the effects of compute (model size, compute available) on performance, while disorganized, is quite interesting.

**Weaknesses:**

There is limited novelty in the approach of removing handcrafted 3D representations – as the authors point out this was done in SRT [1], with a less effective and scalable architecture, as well as in the stereo case [2], which proposes a similar philosophy of input-level inductive biases, and in the multi-view case [3], where the view synthesis branch closely resembles LVSM.  Clarification of the novel elements of this architecture compared to prior work would be great, beyond the discussion of encoder-decoder vs decoder-only.

The presentation is somewhat messy.  The discussion of encoder vs encoder-decover in Section 3.2 takes up a lot of room, but much of it is standard Transformer information; this could be condensed to describe specifically how these concepts relate to the architecture rather than be a general introduction.  Also, there is information about model scale and compute scattered throughout the document, and the model sizes are not in terms of #params but layers.

[1] Sajjadi, Mehdi SM, Henning Meyer, Etienne Pot, Urs Bergmann, Klaus Greff, Noha Radwan, Suhani Vora et al. "Scene representation transformer: Geometry-free novel view synthesis through set-latent scene representations." CVPR 2022

[2] Yifan, Wang, Carl Doersch, Relja Arandjelović, Joao Carreira, and Andrew Zisserman. "Input-level inductive biases for 3d reconstruction." CVPR 2022

[3] Guizilini, Vitor, Igor Vasiljevic, Jiading Fang, Rares Ambrus, Greg Shakhnarovich, Matthew R. Walter, and Adrien Gaidon. "Depth field networks for generalizable multi-view scene representation." ECCV 2022

**Questions:**

Suggestions - clarifying the scaling (model size, compute) discussion (with model parameter counts), and comparison to prior work.
Also if possible it would be good to have the encoder-decoder vs decoder-only comparisons for both scene- and object-level (i.e. Table 2).

---

> ### Author Response · Authors · 2024-11-24
>
> Thank you for your insightful comments and valuable suggestions. We will revise our paper based on your feedback. Here are our responses to your comments:
>
> 1. **Clarification of the novel elements of this architecture compared to prior work**
>
>    Our task is to synthesize novel views from sparse views. Previous SRT shares a similar task as ours, and it also doesn’t have handcrafted 3D representations. However, our two proposed models, Encoder-Decoder LVSM and Decoder-Only LVSM are highly different from SRT. The details are as follows:
>
>     (1). Our encoder-decoder LVSM is similar to SRT (Sajjadi et al., 2022) at a high level – Both SRT and our encoder decoder leverage an encoder to transform input images into a set of 1D tokens that serve as an intermediate latent representation of the 3D scene. A decoder can then render novel view images from this latent representation. However, encoder-decoder LVSM introduces a highly different architecture that significantly improves performance. We have provided related ablation study experiments and a more detailed analysis in our revised paper. (L428-L470, and Table 3). We summarized the key differences as follows:
> 	* (a). We used a simpler and more effective patch-based input image tokenizer, which improves the performance, compared with SRT’s CNN-based tokenizer.
> 	* (b). Our encoder progressively compresses the information from posed input images into a fixed-length set of 1D latent tokens. This design ensures a consistent rendering speed, regardless of the number of input images, as shown in Fig. 6. In contrast, SRT’s latent token size grows linearly with the number of input views, resulting in decreased rendering efficiency.
> 	* (c). The decoder of our encoder-decoder model utilizes pure (bidirectional) self-attention, which enables i) latent tokens to be updated across different transformer layers, which also means the parameters of the decoder are a part of the scene representation; ii) output patch pixels can also attend to other patches for joint updates, ensuring the global awareness of the rendered target image. Prior work SRT (Sajjadi et al., 2022) has a cross-attention-based design for its decoder, which doesn’t support these functions. We experiment with an LVSM variant by adopting SRT’s decoder designs,  which leads to significant performance degradation, as shown in Table 3 of the revised paper.
>
>     (2). Our decoder-only LVSM  further pushes the boundaries of eliminating the inductive bias and bypasses any intermediate representations. It adopts a single-stream transformer to directly convent the input multi-view tokens into target view tokens, treating the view synthesis like a sequence-to-sequence translation task, which is fundamentally different from previous work, including SRT.
>
>     (3). In addition, we have implemented several design choices to make our scalable and effective training happen, including predicting patch pixels instead of ray pixels for more efficient rendering, using better camera pose information imbedding, integrating Flash-Attention, Gradient Checkpointing, and mixed precision training for more efficient training, and using QK-norm for more stable training. The technical designs such as Flash-Attention and Gradient Checkpointing are important for final high-quality results but are often overlooked in the 3D community.
>
>
> 3. **The presentation is messy**
>
>    We sincerely appreciate your suggestions for clarifying our presentation. We have rewritten Sec.3.2 to make it more clear and concise. We have clarified the model size information by providing the model parameter counts in Table 2. We also make the computing information more clear.
>
> 4. **Have Encoder-Decoder VS Decoder-Only on both Object- and Scene-Level**
>
>    We have updated our main paper and provided both the scene- and object-level results in Table 2. The object and scene have observed a similar scaling tendency.

---

> > ### Comment · Reviewer_NhwW · 2024-11-26
> >
> > Though the changes from SRT aren't huge for the encoder-decoder variant, they do allow for more scalable and stable training, and your architecture exploration with the two variants is also quite interesting, I think the novelty has been clarified for me so I've raised my score to 8.

---

> > > ### Author Response · Authors · 2024-11-28
> > >
> > > Thank you for your thoughtful feedback and for acknowledging the clarified novelty and contributions of our work. We're glad the scalability and stability improvements, as well as the architecture exploration, were appreciated. We truly value your feedback and support!

---

### Official Review · Reviewer_X42b · 2024-11-02

**Soundness:** 3
**Presentation:** 3
**Contribution:** 2
**Rating:** 6
**Confidence:** 5

**Summary:**

This paper studies the task of synthesizing novel views from a set of views without explicit 3D representation.
The key idea is to use a transformer architecture to bypass the 3D inductive biases.
To achieve this goal, an encoder-decoder and a decoder-only architectures are proposed to conditional on image and camera pose for novel view synthesis.
Experiments are trained on object-level Objaverse dataset, and scene-level RealEstate10K dataset, and tested on object-level Google Scanned Objects and Amazon Berkeley Objects and scene-level RealEstate10K dataset.
They demonstrate reasonable results on novel view synthesis, outperforming the existing state-of-the-art approaches.

**Strengths:**

### S1 -- Good results on an interesting task
- The task of synthesizing novel views from a set of input views is interesting and very challenging. The proposed method seems to work well on both object-level dataset and scene-level dataset.
- Base on the visual results shown in Figure 3 and 4, the proposed deterministic pipeline also can imagine the new content which is invisible from the input views.

### S2 -- Simple ideas and careful implementations
- There are two main transformer-based architectures.
    - One is an encoder-decoder architecture, which used the transformer encoder to encode all input views into latent space, and then use a query camera pose to get the target view images. This architecture is very similar to SRT, except here the encoder is transformer architecture.
    - The second idea is to design a decoder-only architecture. This reduces the necessary of the one representation from the encoder.
- These two ideas are carefully implemented and validated through ablation studies. In particular, the decoder-only architecture seems quite effective in achieving a set of input views, from 1-10.

### S3 -- Good writing
- The paper is very well written, with clear motivations, sufficient technical explanations and illustrative visualization.

**Weaknesses:**

### W1 --- Significant is not well demonstrated
- The proposed idea is a very specific, minor change to SRT -- basically using a slightly different transformer encoder or decoder to replace the original CNN. Fundamentally, I am not fully convinced that it is even crucial to use only transformer architecture than the CNN-based feature extraction and then do the transform.
- This small change seems to lead to a large improvement on both object-level and scene-level datasets. However, if we train the original SRT on the same dataset with the same computational GPUs, what's the performance?
- A fair comparison to the highly related work (SRT) should be provided. The discussion in L140-145 is also not a very strong claim.
- The decoder-only architecture is very interesting and useful, but some related work like 3DIM, Free3D, CAT3D also used it in the diffusion for a set of views with the Plücker Rays embedding as conditional. Why this architecture is better than these difussion-based methods, which also used the transformer in pixel or latent space?

### W2 --- More results should have been expected
- I expected more visual results on scene-level cross-domain datasets. Figure 4 shows only result on RelEstate10k, which is a relative easy dataset. How about the performance on the scene-level datasets, such as DL3DV, Mip-NeRF 360, or other traditional NeRF Datasets?
- This deterministic model can also provide sharp results for invisible regions, which is quite interesting. However, the authors only highlight them in the object-level results, how about the scene-level results?
- Does the scene-level model perform good for the extrapolation, instead of interpolation?
- In 3DIM, Free3D, SV3D, CAT3D, they used some temporal attention to ensure the consistency of generated images. How do the authors active the 3D consistency in the design? Besides, the multi-view rendered results as in 3DIM and Free3D or reconstructed 3D as in SV3D and CAT3D will make the paper stronger to show 3D consistency and structure, while they want to bypass the 3D representation.
- It will be helpful to provide some visual examples of the failure cases.
- The ablations studies are only related to the different layers of the transformer, and the attention architecture. If the authors argue the large contribution of transformer compared to CNN, a fair ablation should be made to use SRT-related architecture under the same experimental settings.

**Questions:**

- Why this simple architecture performs so good compared to SRT? What's the key contribution, only transformer vs. CNN?
- What's the performance on other scene-level datasets?
- How about the 3D consistency?
- The decoder-only transformer also be used for CAT3D, while they are in latent space for diffusion. If we train the similar architecture, but on diffusion-based architecture, what's the performance? In particular, could we use it to build a large foundational 3D model upon the pre-trained 2D diffusion models? The model not only works well on a special trained dataset, but can be generalised well to arbitrary images.

---

> ### Author Response · Authors · 2024-11-24
>
> Thank you for your insightful comments and valuable suggestions. We will revise our paper based on your feedback. Here are our responses to your comments:
>
> 1. **Difference  between SRT and LVSM**
>
>    We respectfully disagree with the assertion that **“our proposed idea is a very specific, minor change to SRT”**. Prior SRT shares a similar task as ours, and it also doesn’t have handcrafted 3D representations. However, our two proposed models, Encoder-Decoder LVSM and Decoder-Only LVSM are highly different from SRT. Importantly, **we never claimed that the removal of CNNs is the key factor driving the superior performance of our models**. In fact, our models have numerous design differences from SRT, with the removal of CNNs being just one minor component. We apologize for any confusion and will clarify this further as follows:
>
>     (1) Our Encoder-Decoder LVSM is similar to SRT (Sajjadi et al., 2022) at a high level (both use an encoder to transform input images into a set of 1D tokens serving as a latent representation of the 3D scene, which is then decoded to render novel views). However, Encoder-Decoder LVSM introduces a highly different architecture that significantly improves performance. **We have provided related ablation study experiments and a more detailed analysis in our revised paper. (L442-L470, and Table 3)**. We also summarized the key differences as follows:
>     * (a). **Simpler and more effective tokenizer**: We used a simpler and more effective patch-based input image tokenizer, which improves the performance, compared with SRT’s CNN-based tokenizer. (Notably, although removing CNN here brings performance improvement, this change brings less obvious and less significant performance differences ompared with our other design differences, as shown in Table 3 of the revised paper.  We will discuss other design differences later. Therefore, we respectively disagree that our contribution is just **"replace the original CNN"**.  In addition, we would like to point out that SRT employs CNNs only as a input tokenizer and is otherwise pure transformer-based. )
>     * (b). **Progressive compression for fixed-length latent tokens**: Our encoder progressively compresses the information from posed input images into a fixed-length set of 1D latent tokens. This design ensures a consistent rendering speed, regardless of the number of input images, as shown in Fig. 6. In contrast, SRT’s latent token size grows linearly with the number of input views, resulting in decreased rendering efficiency.
>     * (c). **Joint updating of the latent and target patch tokens**: The decoder of our encoder-decoder model utilizes pure (bidirectional) self-attention, which enables i) latent tokens to be updated across different transformer layers, which also means the parameters of the decoder are part of the scene representation; ii) output patch pixels can also attend to other patches for joint updates, ensuring the global awareness of the rendered target image. Prior work SRT (Sajjadi et al., 2022) has a cross-attention-based design for its decoder, which doesn’t support these functions. We experiment with an LVSM variant by adopting SRT’s decoder designs,  which leads to significant performance degradation, as shown in Table 3 of the revised paper (this single change will make PSNR decrease ~3.5dB ). We have also ablated each of the above two functions mentioned individually, showing their effectiveness, and discussed this in detail from L458-469.
>
>   * Our **Decoder-Only** LVSM further pushes the boundaries of eliminating the inductive bias and bypasses any intermediate representations. It adopts a single-stream transformer to directly convent the input multi-view tokens into target view tokens, treating the view synthesis like a sequence-to-sequence translation task, which is fundamentally different from previous work, including SRT.
>
>    * In addition, we have implemented many other design choices to make our scalable and effective training happen, including using plucker rays to better represent the camera information, predicting patch pixels instead of ray pixels for more efficient rendering, integrating Flash-Attention, Gradient Checkpointing, and mixed precision training, for more efficient training, using QK-norm for more stable training, etc. These technical designs are important for final high-quality results.
>
>
> ---
> Unfinished. Please keep reading the comments.

---

> ### Author Response · Authors · 2024-11-25
>
> 2. **Comparison with SRT**
>
>    SRT’s current implementation does not support efficient large-scale training, making it difficult to train SRT on our dataset in a computationally efficient and stable manner.
>    Additionally, as we mentioned in our previous response, the architectural differences between our encoder-decoder model and SRT are not small but significant. They are similar at a high level, but many detailed designs are different, including different input tokenizers, different transformer encoder designs, different latent representations, different transformer decoder attention designs, different camera pose representations, different pixel rendering designs (single ray-based Versus multi-patch-based), etc. Therefore, rather than modifying SRT’s code to work with our dataset, we believe a more meaningful comparison is achieved through ablation studies where we replace our encoder-decoder LVSM model components with SRT’s design. This allows us to isolate the other factors, and better demonstrate and understand the effectiveness of our architectural design choices.
>
>    As shown in the ablation studies that are discussed earlier, our design choices for our encoder-decoder LVSM led to clear performance differences, which we believe provides strong evidence to prove our better performance is not just because of the data we used. We sincerely appreciate your detailed feedback, and we have revised the main paper to include a more detailed discussion and a stronger claim regarding these points in response to your feedback(see L428–L481).
>
>
> 3. **Difference compared with previous Diffusion based methods**
>
>    Our generalizable deterministic methods are fundamentally different from generative models (including diffusion-based models). We have provided a detailed discussion about the difference between our models and the generative models in Appendix A.6 of our revised paper, which has included all the papers you mentioned. If you have further questions or would like additional clarification, we would be happy to provide more detailed responses.
>
> 4. **More scene-level visual results**
>
> 	We appreciate your suggestion of evaluating our model on more complex datasets. In response, we have updated our anonymous project page to include our model's results on the other cross-domain view synthesis datasets, such as MipNeRF360 and LLFF. Please refer to the webpage section titled “Results on Cross-Domain Datasets” for more details. Our model achieves reasonably good results on these datasets.
>
> 	However, we would like to highlight that it is uncommon to use those datasets you mentioned to evaluate sparse-view synthesis models trained on RealEstate10k.  For instance, our scene-level baselines—including PixelSplat (CVPR 2024, Oral, Best Paper Runner-Up), MVSplat (ECCV 2024, Oral), and GS-LRM (ECCV 2024)—are all only validated on RealEstate10k or ACID (which is simpler than RealEstate10k).
>
>    More complex datasets like MipNeRF360 differ significantly from RealEstate10k in terms of data distribution, making them less suitable for evaluating our model. Key differences include:
>       * Camera baseline distances: These model complex datasets involve larger baseline distances between input views.
>       * Camera trajectory distribution: RealEstate10k predominantly features forward or backward camera motion, while those more complex datasets include more different and diverse camera motion trajectories. For example, MipNeRF360 mainly includes 360-degree motion tracks.
>       * Camera intrinsic parameters: Differences include variations in field of view (FOV), aspect ratio, and other intrinsic properties.
>
>    These datasets are more suited for validating view synthesis models trained with denser input views (such as recent LongLRM) and trained on datasets with similar distributions, such as DLV3D.
>
>
> 5. **Failure cases**
>
>    We sincerely appreciate your suggestion, and we have now provided a limitation section in the Appendix. A7. And we have updated our anonymous project page to show failure cases.
>
>
> 6. **Question-related to 3D consistency**
>
>    As demonstrated in the video results on our project webpage, our generalizable deterministic model, trained on large-scale multi-view data, generates high-quality novel view synthesis results with strong view consistency. As we answered previously, unlike probabilistic generative models (e.g., diffusion models), our deterministic approach doesn’t have the inherent view inconsistency issues that are often associated with probabilistic generative frameworks.
>
>    In our revised paper, we have included a detailed discussion on the differences between our model and generative models ,and the view consistency problem. Please refer to Appendix A.6 for more details. And we are happy to discuss if you have further questions.
>
>
> ---
> Unfinished. Please keep reading the comments.

---

> > ### Comment · Reviewer_X42b · 2024-11-27
> > **reply to authors**
> >
> > - Thanks for clarifying the difference between the proposed architecture and SRT. The "Joint updating of the latent and target patch tokens" is reasonable and will lead to better results.
> > - The video is consistency from the project webpage. But the video consistency is different with the 3D consistency. Many video diffusion models also provide consistent video, but the 3D consistency is bad. The authors claimed the Free3D and SV3D also improve the consistency using temporal attention in the diffusion model. However, the SV3D extracts the 3D geometry for all methods to compare the 3D consistency. Hence, it would be better to show the 3D consistency for the rendered novel view, using other tools, like 3D GS or VGGSFM.
> > - The authors only provide the results on rel10k, which is trained on it. Hence, the model may overfit on this dataset. PixelSplat tested the model on ACID, and MVSplat further extended to DTU dataset. The generalisability is very important for the "Large View Synthesis Model".

---

> ### Author Response · Authors · 2024-11-25
>
> 7. **Extrapolation results**
>
>    The anonymous project webpage has already included pose extrapolation video results on object-level data, in which our methods have a good NVS performance.
>
>    We have also updated our anonymous project webpage and provided some extrapolation results for scene data. Given two input images with poses 1 and 2, we generate a novel view rendering video that transitions through poses 0 → 1* → 2* → 3. This means the beginning and end of each video involve pose extrapolation, while the middle segments interpolate between poses 1 and 2. Please visit the anonymous project webpage for visual results.
>
>    As discussed in the limitations section of the revised paper, our models are deterministic. Therefore, similar to all prior deterministic approaches  (e.g., MVSNeRF, IBRNet, SRT, PixelSplat, LRMs, GS-LRM), our models, struggle to produce high-quality results in unseen regions. When the camera moves into unseen areas, the rendering quality degrades, which is often noticeable at the start of the following videos. However, when extrapolation occurs within regions covered by the input views, the rendering quality remains good, as often observed near the end of the following videos. This behavior is also consistent with the observations from single-view synthesis results. We will illustrate this more clearly in the revised paper.
>
>    That said, we found our methods still have better quantitative extrapolation results compared with our baselines. Recent work latentSplat[1] (ECCV 2024) created a testing split of Realestate10k designed for extrapolation testing. We test our extrapolation results on its testing dataset
>    | Method | PSNR↑ | SSIM↑ | LPIPS↓ |
>    | --- | --- | --- | --- |
>    | pixelNeRF [2] | 20.05 | 0.575 | 0.567 |
>    | Du et al. [3] | 21.83 | 0.790 | 0.242 |
>    | pixelSplat [4] | 21.84 | 0.777 | 0.216 |
>    | latenSplat[1] | 22.62 | 0.777 | 0.196 |
>    | Ours Encoder Decoder | 24.86 | 0.827 | 0.164 |
>    | Ours Decoder Only | 26.18 | 0.857 | 0.140 |
>
>    As shown above, our methods perform much better than the baseline in the extrapolation setting. The baseline results here are copied from latentSplat[1] (ECCV 2024)’s Table 2. (We didn’t test MVSplat and GS-LRM here because we have received a lot of reviews and we only have limited time. But we are happy to incorporate them or other related papers suggested by the reviewer in our final revision if the reviewer thinks this is important.)
>
>
>    [1] latentSplat: Autoencoding Variational Gaussians for Fast Generalizable 3D Reconstruction (ECCV 2024)
>
>    [2] pixelNeRF: Neural Radiance Fields from One or Few Images (CVPR 2021)
>
>    [3] Learning to render novel views from wide-baseline stereo pairs (CVPR 2023)
>
>    [4] pixelSplat: 3D Gaussian Splats from Image Pairs for Scalable Generalizable 3D Reconstruction (CVPR 2024)

---

> ### Author Response · Authors · 2024-11-28
>
> Thanks for your further feedback. We explain you left questions as follows:
>
> 1. **More Explanation of the 3D Consistency**
>
>     Our methods are not only visually consistent but also demonstrate strong 3D consistency:
>
>     - We have reported PSNR metrics for both object-level and scene-level novel view synthesis results, as shown in Table 1.  PSNR measures the pixel-level error between our predictions and the ground-truth images, serving as a reliable evaluation metric for 3D consistency, as established in prior NeRF/3DGS methods.
>     - Our results are also consistent enough to reconstruct 3D mesh from it. For example, in the “Results on Cross-Domain Datasets” section of the **updated anonymous webpage**, we demonstrate our method’s ability to render consistent view synthesis results using four sparse input views from cross-domain object-level datasets (e.g., the hotdog example from the NeRF synthetic dataset). Using these synthesized views, we successfully reconstruct a 3D mesh with good quality using common 3D reconstruction methods, such as the recent 2DGS. You can view the reconstructed mesh for the hotdog example here: https://3dviewer.net/#model=https://lvsm-web.github.io/mesh/hotdog.ply . If needed, we are happy to include more results to further support our claims.
> 2. **Cross-Domain Generalization Evaluation**
>     - As explained in our previous reply (Point 4), we have updated our anonymous project page and included our model's results on other cross-domain view synthesis datasets, such as **NeRF-Synthetic**, **MipNeRF360**, and **LLFF**, to show our methods' generalization. Please refer to the webpage section titled *“Results on Cross-Domain Datasets”* for more visual results.
>     - For the quantitative evaluation, to start with, we would like to clarify that **previous work, including pixelSplat and MVSplat**, did not train their models on the RealEstate10k dataset and then evaluate them on ACID. Instead, these methods were **trained directly on ACID and evaluated on ACID.** (In their official repositories, they have released separate checkpoints for Realestate10k and ACID.)
>
>         Our models can outperform prior methods on ACID, even when solely being trained on Realestate10k. The results are as follows: (We test GS-LRM and our methods by ourselves, and report results for other methods based on their paper.)
>
>         | Method |  | ACID |  |
>         | --- | --- | --- | --- |
>         |  | **PSNR ↑** | **SSIM ↑** | **LPIPS ↓** |
>         | pixelNeRF * | 20.97 | 0.547 | 0.533 |
>         | GPNR*| 25.28 | 0.764 | 0.332 |
>         | Du et al. * | 26.88 | 0.799 | 0.218 |
>         | pixelSplat* | 28.27 | 0.843 | 0.146 |
>         | MVSplat* | 28.25 | 0.843 | 0.144 |
>         | GS-LRM | 28.84 | 0.849 | 0.146 |
>         | Ours Encoder-Decoder | 29.05 | 0.846 | 0.159 |
>         | Ours Decoder-Only | **30.44** | **0.869** | **0.126** |
>
>         (Note, ”*” here denotes methods trained with ACID dataset)
>
>         As shown in the table above, our methods—trained only on RealEstate10k—achieve **better or comparable overall performance** on ACID dataset compared to all prior methods, including those explicitly trained on the ACID dataset (e.g., pixelSplat and MVSplat).
>
>         This demonstrates that our methods exhibit **superior performance** and **generalization capabilities**.
>
>
>
> We hope the explanation here addresses your concerns. We sincerely appreciate your valuable efforts and feedback.

---

> > ### Comment · Reviewer_X42b · 2024-11-28
> >
> > Thanks for the updates. All my concerns have been resolved.

---

> > > ### Author Response · Authors · 2024-11-29
> > >
> > > Thank you for your response. We truly value your feedback and are glad to hear that all your concerns have been addressed after our detailed discussion. If you have any additional questions, please don’t hesitate to reach out.
> > >
> > > If everything has been resolved to your satisfaction, we would greatly appreciate it if you could consider adjusting your rating.
> > >
> > > Thank you once again for your time and thoughtful input, which greatly helped enhance this work.

---

### Official Review · Reviewer_jDDo · 2024-11-02

**Soundness:** 4
**Presentation:** 3
**Contribution:** 4
**Rating:** 8
**Confidence:** 5

**Summary:**

This paper introduces two view synthesis models: one with an encoder-decoder architecture and another with a decoder-only design. Both models treat view synthesis as a predictive sequence-to-sequence task, using Plucker embeddings for camera position encoding. Without relying on intermediate 3D structures, the models achieve impressive results across object-level and scene-level view synthesis.

**Strengths:**

The paper is well-motivated and and very well-written, though certain technical details could benefit from additional clarity (outlined below). The visual results are striking, as shown on the authors’ website, and I appreciate the authors provide additional results with limited GPU-hours, making reproduction more feasible for academic labs. Overall, this work is a valuable contribution to view synthesis research.

**Weaknesses:**

1. Related Works. While the paper covers key prior work on 3D representation and few-shot view synthesis, it would benefit from a discussion of generative multi-view methods, especially recent works like Free3D (CVPR 2024, also uses Plucker embedding to encode camera poses) and EscherNet (CVPR 2024, also can be inferenced with varying number of reference/target views). These methods also do not rely on intermediate 3D representations, treating view synthesis as a sequence-to-sequence problem. Adding some proper discussions on how the proposed methods differ from these could provide valuable context for readers, and strength the community focussing on sequence-to-sequence learning type view synthesis.

2. Architecture Design Details.
- Encoder-Decoder Architecture: Could the authors clarify the choice of compressing input tokens into a fixed-length representation with latent tokens? What are the benefits over using uncompressed reference tokens (only with linear complexity)?
- Decoder-Only Architecture: Are attention masks fully bi-directional? It would be interesting to see if introducing asymmetric masking strategies (e.g., limiting specific views to certain tokens) could enhance generalization to different numbers of reference views. Additionally, if attention is fully bi-directional, the distinction between reference and target views seems blurred, as the loss can be computed from all views, (conditioned on all other views)?

3. Experiments

- Unified Model Training: Could the model be trained jointly on both object- and scene-level data instead of using separate models? Demonstrating this capability would advance the model’s utility toward a more general-purpose view synthesis framework.
- Varying Reference Images: The authors suggest the model performs well with varying numbers of reference images. To strengthen this claim, I recommend evaluating on a NeRF-Synthetic dataset instead of GSO in Fig. 5 (similarly as shown in EscherNet), which includes more complex objects in terms of textures and lighting. This would also enable clearer comparisons to other state-of-the-art, scene-specific methods like InstantNGP and 3D Gaussian Splatting that leverage 3D representations.
- Single-Image View Synthesis: Single-image view synthesis is a common use case, so it would be valuable to include a comparison to methods specifically designed for single-image scenarios to showcase the model’s adaptability and strong generalization.
- Plucker Embedding Generalization: Could the authors explore how well the Plucker embeddings generalize to different spatial coordinates? For instance, in scene-level experiments, would generation quality remain consistent if reference/target camera poses are applied with the same camera transformation, such as a 30-degree elevation or a 1-meter translation?

4. Limitations
A dedicated limitations section would help readers identify areas for improvement. Potential limitations include:

- Predictive Modeling: The predictive approach in LVSM may restrict outputs to interpolation, limiting extrapolation capabilities, especially in scene-level tasks (as shown in the website). Maybe a generative model variant trained on larger datasets address this? Can the authors provide some extrapolation results?
- Extreme Reference View Limits: How well does LVSM handle extremes in reference view numbers (e.g., only one view or over 100 views)? This could be an insightful addition, especially if single-view predictions are inconsistent or if performance degrades with many reference views.

I am happy to further raise the score if the authors can adequately address my concerns.

===================

Updated score based on the authors' rebuttal.

**Questions:**

See limitations.

---

> ### Author Response · Authors · 2024-11-24
>
> Thank you for your insightful comments and valuable suggestions. We have revised our paper based on your feedback. Here are our responses to your comments:
>
> 1. **Adding discussions on how the methods differ from generative mult-view methods**
>
>    We sincerely appreciate your suggestion of adding some proper discussions of how our methods differ from the generative methods, including Free3D and EscherNet. We have added a detailed discussion of it in Appendix A6 of our revised paper, and we now have also cited all those papers you mentioned in our “Related Works”.
>
> 2. **Why do we choose to use latent tokens with fixed length**
>
>     Using uncompressed reference tokens does indeed result in linear complexity. However, since we employ bidirectional self-attention in all transformer layers, a linear increase in the transformer's input token size leads to a quadratic growth in computational cost. Consequently, this significantly reduces rendering speed (FPS).
>
> 3. **If the attention is fully bi-directional and if introducing asymmetric masking strategies can help improve performance**
>     * Yes, our current models only use fully bidirectional self-attention blocks for all transformer layers. In L 458-469, we have shown some relevant ablation studies for the masking strategies. The results show that fully bi-directional self-attention achieves the best results across different attention mask designs.
>     * We only predict one target image for each single transformer pass. The target images are attained from the initial target pose tokens, and the loss is only computed on the target image.  We have provided a detailed diagram in our revised papar (Fig.8). Please let us know if you have additional questions.
>
> 4. **Unified Model Training**
>
>     Unified Model Training is an interesting point and we appreciate you mentioning this.
>
>     During the rebuttal period, we tried unified training by mixing the object data and scene data together in the same dataloader. The data loader randomly fetches scene data or object data during training. We also increased the input view number for the scene from the original number 2 to 4, to align with the object data (object data has 4 input images), so that mixed object and scene data can be fed into the the same batch during training. Due to the time limit, we only our decoder-only model under 256 resolution, using the same training configuration (including iteration num, batch size, etc ) as the original scene-level setup. The final model performance is:
>     |  | PSNR | SSIM | LPIPS |
>     | --- | --- | --- | --- |
>     | Realestate10k | 28.81 | 0.894 | 0.107 |
>     | GSO |  30.43 | 0.947 | 0.044 |
>     | ABO | 31.12 | 0.937 | 0.056 |
>
>     Based on the above results, we can see that the model can be jointly trained on both object and scene data, enabling it to handle these two data types within a single framework. This demonstrates its potential as a more general-purpose view synthesis framework capable of managing both object and scene data simultaneously.
>
>     While the jointly trained model exhibits a 0.8–1.3 PSNR performance drop compared to models trained exclusively on either objects or scenes, this is understandable for two key reasons:
>     * Handling both object and scene data in a unified model is inherently more challenging, likely requiring additional model capacity and computational resources.
>     * Due to time constraints, we have not yet explored the optimal training configurations, leaving room for further optimization.
>
>
> 5. **Single-view synthesis comparison**
>
>
>     We want to emphasize that single-view synthesis is not the primary focus of this project. We include some single-view synthesis results solely to demonstrate that our model has a certain level of 3D understanding, such as depth understanding, rather than focusing on pixel-level view interpolation. We will clarify this point more explicitly in the final revised version of the paper.
>
>     To ensure a fair comparison with methods specifically designed for single-image scenarios, we would need to retrain our model using a redesigned training setup to handle single-image inputs. However, this falls outside the scope of our project's focus and cannot be achieved within the limited time frame of the rebuttal period.
>
>     Furthermore, we believe that single-view synthesis inherently involves posing scale ambiguity, making fair quantitative comparisons particularly challenging. Addressing this would require additional time to design a fair evaluation setup, which is also beyond the scope of this project.
>
> ----
> Unfinished. Please keep reading the comments.

---

> > ### Author Response · Authors · 2024-11-24
> >
> > 6. **Test Varying Reference Images on NeRF synthetic data**
> >
> >    We test our decoder-only model on the NeRF-Synthetic dataset with an Extreme view number (from 1 up to 100).
> >    The valuation are conducted in 512 resolution. We report the average PSNR across 8 scenes.
> >
> >     | Metric | 1 Input View(s) | 2 Input View(s) | 3 Input View(s) | 4 Input View(s) | 8 Input View(s) | 16 Input View(s) | 32 Input View(s) | 50 Input View(s) | 100 Input View(s) |
> >     | --- | --- | --- | --- | --- | --- | --- | --- | --- | --- |
> >     | PSNR | 16.40 | 19.91 | 21.51 | 22.24 | 23.44 | 23.63 | 23.68 | 23.73 | 23.75 |
> >
> >     Based on the table, we can see that our model's performance keeps increasing when given more input views (up to 100), without degrading.
> >     We can observe the performance saturates after taking more than 50 views, which is understandable since it has been too much higher than the training input views(4), which can be solved by training our model with varying and larger input view numbers.
> >     We can also observe the performance of NeRF-Synthetic data is overall worse than the ABO and GSO results, which is because (1) NeRF-Synthetic data have more challenging geometry and materials, and they are rendered with more complex lighting conditioning (2) They are rendered with different camera intrinsics, such as FOV, from our training dataset.
> >     Augmenting our training data with better improve our performance. You can find our video results of NeRF-Synthetic data on the anonymous project webpage ("Results on Cross-Domain Dataset." section). (Due to space limit, we only show the results under 4 views.)
> >
> >
> > 7. **Including a Limitation section**
> >
> >     We sincerely appreciate your suggestion, and we have now provided a limitation section in the Appendix. A7.
> >
> > 8. **Plucker Embedding Generalization**
> >
> >     The generalization of the Plücker embedding is highly dependent on the pose distribution of the training data.
> >
> >     In the scene-level experiment, we normalize the scene coordinates during training based on the two input view poses. This ensures that the performance remains unaffected if the same camera transformation is applied to the reference and target views prior to coordinate normalization. However, if the transformation is applied after normalization, performance degrades because the transformed poses fall outside the distribution encountered during training. Training with more diverse pose distribution will help this.
> >
> > ----
> > Unfinished. Please keep reading the comments.

---

> > > ### Author Response · Authors · 2024-11-24
> > >
> > > 9. **Predictive Modeling**
> > >
> > >     The anonymous project webpage has already included pose extrapolation video results on object-level data, in which our methods have a good NVS performance.
> > >
> > >     We have also updated our anonymous project webpage and provided some extrapolation results for scene data. Given two input images with poses 1 and 2, we generate a novel view rendering video that transitions through poses 0 → 1* → 2* → 3. This means the beginning and end of each video involve pose extrapolation, while the middle segments interpolate between poses 1 and 2. Please visit the anonymous project webpage for visual results.
> > >
> > >     As discussed in the limitations section of the revised paper, our models are deterministic. Therefore, similar to all prior deterministic approaches  (e.g., MVSNeRF, IBRNet, SRT, PixelSplat, LRMs, GS-LRM), our models, struggle to produce high-quality results in unseen regions. When the camera moves into unseen areas, the rendering quality degrades, which is often noticeable at the start of the following videos. However, when extrapolation occurs within regions covered by the input views, the rendering quality remains good, as often observed near the end of the following videos. This behavior is also consistent with the observations from single-view synthesis results. We will illustrate this more clearly in the revised paper.
> > >
> > >     That said, we found our methods still have better quantitative extrapolation results compared with our baselines. Recent work **latentSplat[1]** (ECCV 2024) created a **testing split of Realestate10k designed for extrapolation testing**. We test our extrapolation results on its testing dataset
> > >
> > >     | Method | PSNR↑ | SSIM↑ | LPIPS↓ |
> > >     | --- | --- | --- | --- |
> > >     | pixelNeRF [2] | 20.05 | 0.575 | 0.567 |
> > >     | Du et al. [3] | 21.83 | 0.790 | 0.242 |
> > >     | pixelSplat [4] | 21.84 | 0.777 | 0.216 |
> > >     | latenSplat[1] | 22.62 | 0.777 | 0.196 |
> > >     | Ours Encoder Decoder | 24.86 | 0.827 | 0.164 |
> > >     | Ours Decoder Only | 26.18 | 0.857 | 0.140 |
> > >
> > >     As shown above, our methods perform much better than the baseline in the extrapolation setting. The baseline results here are copied from latentSplat[1] (ECCV 2024)’s Table 2. (We didn’t test MVSplat and GS-LRM here because we have received a lot of reviews and we only have limited time. But we are happy to incorporate them or other related papers suggested by the reviewer in our final revision if the reviewer thinks this is important.)
> > >
> > >
> > >     [1] latentSplat: Autoencoding Variational Gaussians for Fast Generalizable 3D Reconstruction (ECCV 2024)
> > >
> > >     [2] pixelNeRF: Neural Radiance Fields from One or Few Images (CVPR 2021)
> > >
> > >     [3] Learning to render novel views from wide-baseline stereo pairs (CVPR 2023)
> > >
> > >     [4] pixelSplat: 3D Gaussian Splats from Image Pairs for Scalable Generalizable 3D Reconstruction (CVPR 2024)

---

> > ### Comment · Reviewer_jDDo · 2024-11-24
> > **Response**
> >
> > Thanks to the authors for the detailed response.
> >
> > 1. The concern is resolved.
> >
> > 2. Without additional context, encoder-decoder architectures typically assume to use cross-attention. Based on the newly attached Fig 9, the paper seems to be using spatially concated self-attention -- which is the key design element to ensure good performance based on ablation. I would suggest to highlight this in the main paper. e.g. just adding a parathesis or one sentence saying like "We employ spatial concatenation to combine both latent inputs and target outputs in a single token sequence and apply self-attention in both encoder-decoder and decoder-only architectures."
> >
> > 3. The concern is resolved.
> >
> > 4. The concern is resolved.
> >
> > 5. The concern is resolved.
> >
> > 6. The concern is resolved.
> >
> > 7. The concern is resolved.
> >
> > 8. The concern is resolved.
> >
> > 9. Are views in Objaverse sampled in an object-centric manner on a sphere with a fixed view radius? If so, I wouldn't consider results in Objaverse to be pose extrapolation? But anyway, the additional results in scene-level synthesis still look outstanding.
> >
> > As nearly all my concerns have been resolved, I am happy to raise my score to 8.

---

> ### Author Response · Authors · 2024-11-25
>
> Thanks for your further feedback.
>
> For point 2, we understand the naming of the decoder can be a little confusing for some readers. In our original paper, we used a separate paragraph in Sec. 3.2 to explain this. During the rebuttal period, we received some feedback from another reviewer for this issue, and we revised our paper based on the reviewer's suggestion and now mention the decoder naming in L239 and Appendix A1. We sincerely appreciate your suggestion and will further clarify this point in the final paper based on your suggestion.
>
> For point 9, the input views are sampled in an object-centric manner on a sphere with a fixed view radius. But the rendered novel view synthesis video results are sampled with varying radius (from 1.8 to 3.2), which makes us think the object results are extrapolation.

---

> > ### Comment · Reviewer_jDDo · 2024-11-25
> >
> > Thanks for the updates. All my concerns have now been resolved.

---

### Official Review · Reviewer_h9gs · 2024-11-04

**Soundness:** 4
**Presentation:** 4
**Contribution:** 4
**Rating:** 8
**Confidence:** 5

**Summary:**

This work presented the Large View Synthesis Model (LVSM), which aims to achieve novel view synthesis (NVS) via a pure transformer architecture, bypassing the need for additional 3D inductive bias. In particular, LVSM explored an encoder-decoder model design and a decoder-only one, where the former is more efficient in inference with a more compact learned latent space, and the latter is more effective and scalable regarding visual quality. Experiments of object-level datasets and scene-level datasets demonstrate the superiority of the introduced LVSM.

**Strengths:**

* The idea of achieving high-quality photorealistic NVS with minimal 3D inductive bias is brave. It is also impressive that LVSM implements this brave idea with a straightforward yet effective pure Transformer-based architecture.
* Experiments on several benchmarks demonstrate the effectiveness of the introduced LVSM
* The paper is well structured, and it is easy to follow.

**Weaknesses:**

* More discussion with Scene Representation Transformer (SRT) [Sajjadi et. al, CVPR 22]. LVSM seems to be a ‘reimplementation’ of SRT with more recent modules, which significantly limits the novelty of LVSM. The discussions in L141-L146 cannot convince me about the key contribution of LVSM. A more thorough analysis is suggested below.
  * The introduction should clearly reveal the similarities and differences between SRT and LVSM. The motivation (minimal 3D inductive bias) and architecture (encoder-decoder) of LVSM are similar to SRT, and it seems that the key difference is that LVSM adopts more advanced modules from LRM.
  * Different observations of ‘decoder-only’ architecture between LVSM and SRT.  SRT also explores the ‘decoder-only’ designs (see Sec. 4.3 “No Encoder Transformer” in SRT), which shows that ‘decoder-only’ performs worse than its ‘encoder-decoder’ counterparts. This observation happens to be contradicted to that of LVSM. It would be interesting to provide further analysis about what leads to this different conclusion despite similar settings.
  * More analysis is needed to understand why LVSM achieves good quality on NVS.  It is hard to understand why LVSM gets much cleaner images while SRT gets only blurry ones. Is it because LVSM is trained with more data? Besides, it would be beneficial to visualise the attention of LVSM (similar to Fig. 5 in SRT).
  * Noisy poses or unknown poses settings on LVSM. SRT achieves reasonably good results when the camera poses are noisy or even unknown (Fig. 4 in SRT). How does LVSM perform under similar noisy pose or pose-free settings?


* Assessing the geometrical accuracy. For the object-level data, it would be better to reconstruct the 3D mesh using the rendered novel views, similar to Fig. 5 in latentSplat [Wewer et al., ECCV 24]. For the scene-level data, e.g., the single-view case in Fig. 1, it would be better to show the error map between the rendered and ground truth views (similar to Fig.6 in MVSplat [Chen et al. ECCV 24]), which makes it easier to understand how well the rendered views align with the given camera poses.


* Performance on more complex datasets. L527 claims that LVSM is not simply doing view interpolation. Since the results are mainly shown on simple data, e.g., RealEstate10K with zoom-in / zoom-out trajectories, it cannot justify this claim. It would be better to show some results on more complex datasets, e.g., MipNeRF360, Tanks and Temples.

**Questions:**

Kindly refer to [Weaknesses]

---

> ### Author Response · Authors · 2024-11-24
>
> Thank you for your insightful comments and valuable suggestions. We have revised our paper based on your feedback. Here are our responses to your comments:
>
> 1. **LVSM is not a ‘reimplementation’ of SRT**
>
>    We respectfully disagree that “LVSM seems to be a ‘reimplementation’ of SRT with more recent modules”.
>    It’s true that both SRT and LVSM share the same motivation – removing 3D inductive bias, which was first explored in Light field networks[1](NeurIPS 2021),  but our models (Encoder-Decoder LVSM and Decoder-Only LVSM) have highly different architecture designs from SRT. To illustrate,  Encoder-Decoder LVSM and SRT both have encoder-decoder designs, but this only makes them similar at a high level and LVSM has a lot of different architecture designs, which significantly improve our performance. (More details will be discussed in point 2 of this rebuttal reply). In addition, our Decoder-Only LVSM achieves better performance than our Encoder-Decoder and it adopts a single-stream transformer to directly convent the input multi-view tokens into target view tokens, treating the view synthesis like a sequence-to-sequence translation task, which is fundamentally different from previous work, including SRT.
>
>     [1]Light field networks: Neural scene representations with single-evaluation rendering
>
> 2. **More Analysis regarding why we have better performance**
>
>     (1) Our Encoder-Decoder LVSM is similar to SRT (Sajjadi et al., 2022) at a high level –both use an encoder to transform input images into a set of 1D tokens serving as a latent representation of the 3D scene, which is then decoded to render novel views. However, Encoder-Decoder LVSM introduces a highly different architecture that significantly improves performance. We appreciate your suggestion of making a more thorough analysis in the main paper. We have revised our main paper and provided related ablation study experiments with a more detailed analysis (L442-L470, and Table 3). We also summarized the key differences as follows:
>       * (a). Simpler and more effective tokenizer: We used a simpler and more effective patch-based input image tokenizer, which improves the performance and also makes the training more stable.
>       * (b). Progressive compression for fixed-length latent tokens: Our encoder progressively compresses the information from posed input images into a fixed-length set of 1D latent tokens. This design ensures a consistent rendering speed, regardless of the number of input images, as shown in Fig. 6. In contrast, SRT’s latent token size grows linearly with the number of input views, resulting in decreased rendering efficiency.
>       * (c). Joint updating of the latent and target patch tokens: The decoder of our encoder-decoder model utilizes pure (bidirectional) self-attention, which enables i) latent tokens to be updated across different transformer layers, which also means the parameters of the decoder are a part of the scene representation; ii) output patch pixels can also attend to other patches for joint updates, ensuring the global awareness of the rendered target image. Prior work SRT (Sajjadi et al., 2022) has a cross-attention-based design for its decoder, which doesn’t support these functions. We experiment with an LVSM variant by adopting SRT’s decoder designs,  which leads to significant performance degradation, as shown in Table 3 of the revised paper (this single change will make PSNR will decrease ~3.5dB ). We have also ablated each of the above functions individually, showing their effectiveness, and discussed this in detail from L458-469.
>
>     (2) Our Decoder-Only LVSM  further pushes the boundaries of eliminating the inductive bias and bypasses any intermediate representations. It adopts a single-stream transformer to directly convent the input multi-view tokens into target view tokens, treating the view synthesis like a sequence-to-sequence translation task, which is fundamentally different from previous work, including SRT. We will discuss why decoder-only has better performance in point 3 of this reply.
>
>
>     (3) In addition, we have implemented many other design choices to make our scalable and effective training happen, including using plucker rays to better represent the camera information, predicting patch pixels instead of ray pixels for more efficient rendering, integrating Flash-Attention, Gradient Checkpointing, and mixed precision training, for more efficient training, using QK-norm for more stable training, etc. These technical designs are important for final high-quality results.
>
> ---
> Unfinished. Please keep reading the comments.

---

> ### Author Response · Authors · 2024-11-24
>
> 3. **Explanation for the Different observations of ‘decoder-only’ architecture**
>
>    The superior performance of our decoder-only model compared to the encoder-decoder model aligns with our observation in Table 2 of the main paper: reducing the number of encoder layers while increasing decoder layers improves the performance of the Encoder-Decoder LVSM.
>
> 	This contradictes with SRT’s observation simply because LVSM and SRT have highly different decoder architectures,(which again proves LVSM is not a reimplementation of SRT). To illustrate, we use a (bidirectional) self-attention block for each decoder layer while SRT uses pure cross-attention. (Further details are discussed in point 2 of this reply.) Our decoder block design achieves superior performance, which makes the decoder-only achieve the best results according to Table 1.
>
> 4. **LVSM Performance Under Unknown Pose Settings**
>
>    While we did not explicitly train LVSM under unknown pose settings as done by SRT, the model demonstrates promising single-view synthesis capabilities when trained solely on multi-view data, as evidenced by the video results available on our anonymous webpage. This suggests that LVSM has the potential to handle tasks without explicit pose information, as single-view synthesis inherently falls into this category—since a single input view can correspond to an arbitrary pose.
>
>    We believe that training LVSM specifically on unposed/single-image inputs could further enhance its performance in such scenarios. However, due to time constraints, we were unable to explore these experiments at this time, leaving it as a promising future work.
>
> 5. **Assessing the geometry accuracy**
>
>    We sincerely appreciate your suggestions for assessing geometrical accuracy. Below is our detailed response:
>
> 	* Our model can be used to reconstruct 3D mesh from sparse views. In the  “Results on Cross-Domain Datasets” section of the updated anonymous webpage, we show our methods can render consistent view synthesis results from cross-domain object-level datasets (such as the hotdog example from the NeRF synthetic dataset). We can then reconstruct a 3D mesh from the view synthesis video using 2DGS(same as latentSplat you mentioned), the result can be seen here: https://3dviewer.net/#model=https://lvsm-web.github.io/mesh/hotdog.ply . We didn’t show mesh reconstruction results in the paper since we focused on view synthesis. However, we are happy to include more results in our final revised paper if you think this is important.
> 	*  For the single-view case, we want to emphasize that single-view synthesis is not the primary focus of this project. We include some single-view synthesis results solely to demonstrate that our model has a certain level of 3D understanding, such as depth understanding, rather than focusing on pixel-level view interpolation. We will clarify this point more explicitly in the final revised version of the paper. Furthermore, we believe that single-view synthesis inherently involves scene scale ambiguity, making fair quantitative comparisons and error map visualization particularly challenging. The MVSplat work you mentioned shows an error map under two input view settings, which don’t involve scene scale ambiguity
>
> ---
> Unfinished. Please keep reading the comments.

---

> > ### Author Response · Authors · 2024-11-24
> >
> > 6. **Performance on more complex datasets**
> >
> >    In L527, we stated that “LVSM is not simply performing view interpolation” by demonstrating its capability to generate reasonable results under single-view synthesis. We believe this is a fair claim because, when provided with only a single view, the model lacks additional views for interpolation. To produce the reasonable results showcased on our webpage, the model must possess some form of 3D understanding—such as implicit scene depth perception—derived from the single view.
> >
> >    We appreciate your suggestion to evaluate our model on more complex datasets. In response, we have updated our anonymous project page to include results on widely used view synthesis datasets, such as MipNeRF360 and LLFF. Please refer to the webpage section titled ***“Results on Cross-Domain Datasets”*** for more details. Our model achieves reasonably good results on these datasets.
> >
> > 	However, we would like to highlight that it is uncommon to evaluate sparse-view synthesis models trained on RealEstate10k on such datasets. For instance, our scene-level baselines—including PixelSplat (CVPR 2024, Oral, Best Paper Runner-Up), MVSplat (ECCV 2024, Oral), and GS-LRM (ECCV 2024)—are typically validated only on RealEstate10k or ACID (which is simpler than RealEstate10k).
> >
> >    More complex datasets like MipNeRF360 differ significantly from RealEstate10k in terms of data distribution, making them unsuitable for validating our model. Key differences include:
> >    * Camera baseline distances: These model complex datasets involve larger baseline distances between input views.
> >    * Camera trajectory distribution: RealEstate10k predominantly features forward or backward camera motion, while those more complex datasets include more different and diverse camera motion trajectories. For example, MipNeRF360 mainly includes 360-degree motion tracks.
> >    * Camera intrinsic parameters: Differences include variations in field of view (FOV), aspect ratio, and other intrinsic properties.
> >
> >    These datasets are more suited for validating view synthesis models trained with denser input views (such as recent LongLRM) and trained on datasets with similar distributions, such as DLV3D.

---

> > > ### Comment · Reviewer_h9gs · 2024-11-28
> > >
> > > I want to express my gratitude for the authors' thorough responses. They addressed all my previous concerns. I believe LVSM will provide a new perspective to the NVS community. I'm happy to raise my rating to 8. Very nice work, well done.

---

> > > > ### Author Response · Authors · 2024-11-28
> > > >
> > > > Thank you for your thoughtful feedback and support!
> > > > We're glad our responses addressed your concerns and appreciate your recognition of LVSM's potential in the NVS community.

---

### Author Response · Authors · 2024-11-25
**We have updated the revised PDF and responsed to every reviewer individually**

Dear Reviewers,

This work has received a lot of reviews with detailed comments. We deeply appreciate the time and effort you have dedicated to providing detailed and insightful feedback on our work. Your comments have been valuable in improving the quality of our paper.

In response, we have carefully addressed all your concerns individually and made revisions to both our main paper and the anonymous project page. The primary revisions include, but are not limited to, the following:

* We have updated Table 3 and Section 4.4 to provide a more detailed analysis of the effectiveness of our model designs and the difference between our approach and prior work SRT.
* We have updated Table 2 to provide more information and results of our model size scaling experiments
* We have included a detailed model architecture diagram in Figure 8 to help the reader better understand our model designs in details.
* We have provided a new section A6 in our appendix to discuss the difference between our models and prior generative models
* We have provided a new section A7 in our appendix to discuss our limitation
* We have revised our Section 3.2 to improve our presentation.

We sincerely thank you again for your thoughtful reviews and look forward to receiving your further feedback.

---

### Meta-Review · Area_Chair_6en8 · 2024-12-17

**Metareview:**

The paper introduces LVSM, a transformer-based approach for novel view synthesis that effectively minimizes 3D inductive biases while achieving state-of-the-art performance across object- and scene-level datasets. Reviewers praised the impressive visual and quantitative results, the thoughtful architectural design of both encoder-decoder and decoder-only variants, and the extensive ablation studies that clarify the contributions of key model components. The model's scalability, efficient training on limited computational resources, and generalizability to varying input views were also highlighted as significant strengths, making this work a valuable contribution to the novel view synthesis community.

**Additional Comments On Reviewer Discussion:**

During the rebuttal period, the authors clarified key differences between LVSM and prior methods like SRT, emphasizing architectural innovations such as progressive latent compression, bidirectional self-attention, and the decoder-only transformer design, supported by ablation studies. They addressed concerns about generalization and complexity by providing additional results on datasets like MipNeRF360, LLFF, and ACID, demonstrating LVSM's strong performance and cross-domain generalizability. The authors also improved clarity with detailed architectural diagrams, added failure cases and limitations, and showcased pose extrapolation results, resolving reviewer concerns and strengthening the paper’s contributions.

---

### Decision · Program_Chairs · 2025-01-22

Accept (Oral)